# Near-Optimal Linear Regression under Distribution Shift

## Abstract

Transfer learning is an essential technique when sufficient data comes from the source domain, while no or scarce data is from the target domain. We develop estimators that achieve minimax linear risk for linear regression problems under the distribution shift. Our algorithms cover different kinds of settings with covariate shift or model shift. We also consider when data are generating from either linear or general nonlinear models. We show that affine minimax rules are within an absolute constant of the minimax risk even among nonlinear rules for various source/target distributions.

## 1 Introduction

The success of machine learning crucially relies on the availability of labeled data. The data labeling process usually requires much human labor and can be very expensive and time-consuming, especially for large datasets like ImageNet (Deng et al., 2009). On the other hand, models trained on one dataset, despite performing well on test data from the same distribution they are trained on, are often sensitive to *distribution shifts*, i.e., they do not adapt well to related but different distributions. Even small distributional shift can result in substantial performance degradation (Recht et al., 2018; Lu et al., 2020).

Transfer learning has been an essential paradigm to tackle the challenges associated with insufficient labeled data (Pan & Yang, 2009; Weiss et al., 2016; Long et al., 2017). The main idea is to make use of a *source domain* with a lot of labeled data (e.g. ImageNet), and to try to learn a model that performs well on our *target domain* (e.g. medical images) where few or no labels are available. Despite the lack of labeled data, we may still use unlabeled data from the target domain, which are usually much easier to obtain and can provide helpful information about the target domain. Although this approach has been integral to many applications, many fundamental questions are left open even in very basic settings.

In this work, we focus on the setting of *linear regression under distribution shift* and ask the fundamental question of how to optimally learn a linear model for a target domain, using labeled data from a source domain and unlabeled data (and possibly some labeled data) from the target domain. For various settings, including covariate shift (i.e., when $p(\boldsymbol{x})$ changes) and model shift (i.e., when $p(y|\boldsymbol{x})$ changes), we develop estimators that achieve *near minimax risk* (up to universal constant factors) among all linear estimation rules. Here linear estimators refer to all estimators that depend linearly on the label vector; these include almost all popular estimators known in linear regression, such as ridge regression and its variants. When the input covariances in source and target domains commute, we prove that our estimators achieve near minimax risk among all possible estimators.

A key insight from our results is that, when covariate shift is present, we need to apply data-dependent regularization that adapts to changes in the input distribution. For linear regression, this can be given by the input covariances of source and target tasks, which can be estimated using unlabeled data. Our experiments verify that our estimator has significant improvement over ridge regression and similar heuristics.

## 1.1 RELATED WORK

**Different types of distribution shift** are introduced in (Storkey, 2009; Quionero-Candela et al., 2009). Specifically, covariate shift occurs when the marginal distribution on $P(X)$ changes from source to target domain (Shimodaira, 2000; Huang et al., 2007). Wang et al. (2014); Wang & Schneider (2015) tackle model shift ($P(Y|X)$) provided the change is smooth as a function of $X$. Sun et al. (2011) design a two-stage reweighting method based on both covariate shift and model shift. Other methods like the change of representation, adaptation through prior, and instance pruning are proposed in (Jiang & Zhai, 2007). In this work, we focus on the above two kinds of distribution shift. For modeling target shift ($P(Y)$) and conditional shift ($P(X|Y)$), Zhang et al. (2013) exploits the benefit of multi-layer adaptation by some location-scale transformation on $X$.

**Transfer learning/domain adaptation** are sub-fields within machine learning to cope with distribution shift. A variety of prior work roughly falls into the following categories. 1) Importance-reweighting is mostly used in the covariate shift. (Shimodaira, 2000; Huang et al., 2007; Cortes et al., 2010); 2) One fruitful line of work focuses on exploring robust/causal features or domain-invariant representations through invariant risk minimization (Arjovsky et al., 2019), distributional robust minimization (Sagawa et al., 2019), human annotation (Srivastava et al., 2020), adversarial training (Long et al., 2017; Ganin et al., 2016), or by minimizing domain discrepancy measured by some distance metric (Pan et al., 2010; Long et al., 2013; Baktashmotlagh et al., 2013; Gong et al., 2013; Zhang et al., 2013; Wang & Schneider, 2014) ; 3) Several approaches seek gradual domain adaptation (Gopalan et al., 2011; Gong et al., 2012; Glorot et al., 2011; Kumar et al., 2020) through self-training or a gradual change in the training distribution.

**Near minimax estimations** are introduced in Donoho (1994) for linear regression problems with Gaussian noise. For a more general setting, Juditsky et al. (2009) estimate the linear functional using convex programming. Blaker (2000) compares ridge regression with a minimax linear estimator under weighted squared error. Kalan et al. (2020) considers a setting similar to this work of minimax estimator under distribution shift, but focuses on computing the lower bound for linear and one-hidden-layer neural network under distribution shift. A few more interesting results are derived on the generalization lower bound for distribution shift under various settings (David et al., 2010; Hanneke & Kpotufe, 2019; Ben-David et al., 2010; Zhao et al., 2019).

## 2 PRELIMINARY

We formalize the setting considered in this paper for transfer learning under the distribution shift.

**Notation and setup.** Let $p_S(\boldsymbol{x})$ and $p_T(\boldsymbol{x})$ be the marginal distribution for $\boldsymbol{x}$ in source and target domain. The associated covariance matrices are $\Sigma_S$, and $\Sigma_T$. We assume to have sufficient unlabeled data to estimate $\Sigma_T$ accurately. We observe $n_S, n_T$ labeled samples from source and target domain. Data is scarce in target domain: $n_S \gg n_T$ and $n_T$ can be 0. Specifically, $X_S = [\boldsymbol{x}_1^\top | \boldsymbol{x}_2^\top | \cdots | \boldsymbol{x}_{n_S}^\top]^\top \in \mathbb{R}^{n \times d}$, with $\boldsymbol{x}_i, i \in [n_S]$ drawn from $p_S$, noise $\boldsymbol{z} = [z_1, z_2, \cdots z_{n_S}]^\top, z_i \sim \mathcal{N}(0, \sigma^2)$. $\boldsymbol{y}_S = [y_1, y_2, \cdots, y_{n_S}]^\top \in \mathbb{R}^{n_S}$, with each $y_i = f^*(x_i) + z_i$ ($X_T \in \mathbb{R}^{n_T \times d}$ and

$\boldsymbol{y}_T \in \mathbb{R}^{n_T}$ are similarly defined). Denote by $\hat{\Sigma}_S = X_S^\top X_S / n_S$ the empirical covariance matrix (Throughout the paper we assume data is centered: $\mathbb{E}_{p_S}[\boldsymbol{x}] = \mathbb{E}_{\boldsymbol{p}_T}[\boldsymbol{x}] = 0$). The positive part of a number is denoted by $(x)_+$. We consider both linear ($f^*(\boldsymbol{x}) = \boldsymbol{x}^\top \boldsymbol{\beta}^*$) and general nonlinear ground truth models. When the optimal linear model changes from source to domain we add a subscript for distinction, i.e. , $\boldsymbol{\beta}_S^*$ and $\boldsymbol{\beta}_T^*$. We use bold ($\boldsymbol{x}$) symbols for vectors, lower case letter ($x$) for scalars and capital letter ($A$) for matrices.

**Minimax (linear) risk.** In this work, we focus on designing linear estimators $\hat{\boldsymbol{\beta}} = A\boldsymbol{y}_S$[1] for parameter $\boldsymbol{\beta}^* \in \mathcal{B}$. Our estimator is evaluated by the excess risk on target domain, with the worst case $\boldsymbol{\beta}^*$ in some set $\mathcal{B}$: $L_\mathcal{B}(\hat{\boldsymbol{\beta}}) = \max_{\boldsymbol{\beta}^* \in \mathcal{B}} \mathbb{E}_{\boldsymbol{y}_S} \|\Sigma_T^{1/2}(\hat{\boldsymbol{\beta}}(\boldsymbol{y}_S) - \boldsymbol{\beta}^*)\|^2$. Minimax linear risk and minimax risk among all estimators are respectively defined as:

$$R_L(\mathcal{B}) \equiv \min_{\hat{\boldsymbol{\beta}} \text{ linear in } \boldsymbol{y}_S} L_\mathcal{B}(\hat{\boldsymbol{\beta}}); \quad R_N(\mathcal{B}) \equiv \min_{\hat{\boldsymbol{\beta}}} L_\mathcal{B}(\hat{\boldsymbol{\beta}}).$$

The subscript "N" or "L" is a mnemonic for "non-linear" or "linear" estimators. $R_N$ is the optimal risk with no restriction placed on the class of estimators. $R_L$ only considers the linear function class for $\hat{\boldsymbol{\beta}}$. Minimax linear estimator and minimax estimator are the estimators that respectively attain $R_L$ and $R_N$ within universal multiplicative constants. Normally we only consider $\mathcal{B} = \{\boldsymbol{\beta} | \|\boldsymbol{\beta}\|_2 \le r\}$. When there is no ambiguity, we simplify $\hat{\boldsymbol{\beta}}(\boldsymbol{y}_S)$ by $\hat{\boldsymbol{\beta}}$.

**Our meta-algorithm.** Our paper considers different settings with distribution shift. Our methods are unified under the following meta-algorithm:

Step 1: Find an unbiased sufficient statistic $\hat{\boldsymbol{\beta}}_{\text{SS}}$[2] for the unknown parameter.
Step 2: Find $\hat{\boldsymbol{\beta}}_{\text{MM}}$, a linear operator applied to $\hat{\boldsymbol{\beta}}_{\text{SS}}$ that minimizes $L_\mathcal{B}(\hat{\boldsymbol{\beta}}_{\text{MM}})$.

For each setting, we will show that $\hat{\boldsymbol{\beta}}_{\text{MM}}$ achieves linear minimax risk $R_L$ (asymptotically or in fixed design). Furthermore, under some conditions, the minimax risk $R_N$ is uniformly lower bounded by a universal constant times $L_\mathcal{B}(\hat{\boldsymbol{\beta}}_{\text{MM}})$.

**Outline.** In the sections below, we tackle the problem in different settings. In Section 3 we design algorithms with only covariate shift: 1) $n_T = 0$ and $f^*(\boldsymbol{x})$ is linear (Section 3.1); 2) $n_T = 0$ and $f^*(\boldsymbol{x})$ is a general nonlinear function (Section 3.2); 3) $n_T > 0$ and $f^*$ is linear (Section 3.3). Finally, we cope with the model shift for linear models ($\boldsymbol{\beta}_S^* \ne \boldsymbol{\beta}_T^*$) in Section 4.

## 3 MINIMAX ESTIMATOR WITH COVARIATE SHIFT

In this section, we consider the setting with only covariate shift. That is, only $\Sigma_S$ (marginal distribution $p_S(\boldsymbol{x})$) changes to $\Sigma_T$ ($p_T(\boldsymbol{x})$), but $f^* = \mathbb{E}[y|\boldsymbol{x}]$ (conditional distribution $p(y|\boldsymbol{x})$) is shared. We first consider the case when $f^*$ is a linear map: $\boldsymbol{x} \to \boldsymbol{x}^\top \boldsymbol{\beta}^*$ and then consider the problem with approximation power.

---

[1] $A \in \mathbb{R}^{d \times n}$ may depend in an arbitrary way on $X_S, n_S$, or $\Sigma_T$. The estimator is linear in the observation $y_S$.
[2] With samples $\boldsymbol{y}_S$, a statistic $t = T(\boldsymbol{y}_S)$ is sufficient for the underlying parameter $\boldsymbol{\beta}^*$ if the conditional probability distribution of the data $\boldsymbol{y}_S$, given the statistic $t = T(\boldsymbol{y}_S)$, does not depend on the parameter $\boldsymbol{\beta}^*$.

### 3.1 COVARIATE SHIFT WITH LINEAR MODELS

We observe $n_S$ samples from source domain: $\boldsymbol{y}_S = X_S\boldsymbol{\beta}^* + \boldsymbol{z}, \boldsymbol{z} \sim \mathcal{N}(0, \sigma^2 I)$ and no labeled samples from the target domain. Our goal is to find the minimax linear estimator $\hat{\boldsymbol{\beta}}_{\text{MM}}(\boldsymbol{y}_S) = A\boldsymbol{y}_S$ with some linear mapping $A$ that attains $R_L(\mathcal{B})$.

Following our meta-algorithm, let $\hat{\boldsymbol{\beta}}_{\text{SS}} = \frac{1}{n_S}\hat{\Sigma}_S^{-1}X_S^\top \boldsymbol{y}_S$[3] be an unbiased sufficient statistic for $\boldsymbol{\beta}^*$:

$$\hat{\boldsymbol{\beta}}_{\text{SS}} = \frac{1}{n_S}\hat{\Sigma}_S^{-1}X_S^\top \boldsymbol{y}_S = \frac{1}{n_S}\hat{\Sigma}_S^{-1}X_S^\top X_S\boldsymbol{\beta}^* + \frac{1}{n_S}\hat{\Sigma}_S^{-1}X_S^\top \boldsymbol{z}.$$

$$= \boldsymbol{\beta}^* + \frac{1}{n_S}\hat{\Sigma}_S^{-1}X_S^\top \boldsymbol{z} \sim \mathcal{N}\left(\boldsymbol{\beta}^*, \frac{\sigma^2}{n_S}\hat{\Sigma}_S^{-1}\right). \tag{1}$$

The fact that $\hat{\boldsymbol{\beta}}_S S(\boldsymbol{y}_S)$ is a sufficient statistic is proven in Claim 3.7 for a more general case, using the Fisher-Neyman factorization theorem. Here we consider $X_S$ as fixed values, and randomness only comes from noise $\boldsymbol{z}$. We prove that the minimax linear estimator is of the form $\hat{\boldsymbol{\beta}}_{\text{MM}} = C\hat{\boldsymbol{\beta}}_{\text{SS}}$ and then design algorithms that calculate the optimal $C$.

**Claim 3.1.** *The minimax linear estimator is of the form $\hat{\boldsymbol{\beta}}_{MM} = C\hat{\boldsymbol{\beta}}_{SS}$ for some $C \in \mathbb{R}^{d\times d}$.*

**Warm-up: commutative covariance matrices.** In order to derive the minimax linear estimator, we first consider the simple case when $\Sigma_T$ and $\hat{\Sigma}_S$ are simultaneously diagonalizable. We apply Pinsker's Theorem (Johnstone, 2011) and get:

**Theorem 3.2** (Linear Minimax Risk with Covariate Shift). *Suppose the observations follow sequence model $\boldsymbol{y}_S = X_S\boldsymbol{\beta}^* + \boldsymbol{z}, \boldsymbol{z} \sim \mathcal{N}(0, \sigma^2 I_n)$. If $\Sigma_T = U\mathrm{diag}(\boldsymbol{t})U^\top$ and $\hat{\Sigma}_S \equiv X_S^\top X_S/n_S = U\mathrm{diag}(\boldsymbol{s})U^\top$, then the minimax linear risk*

$$R_L(\mathcal{B}) \equiv \min_{\hat{\boldsymbol{\beta}}=A\boldsymbol{y}_S}\max_{\boldsymbol{\beta}^*\in\mathcal{B}}\mathbb{E}\,\|\Sigma_T^{1/2}(\hat{\boldsymbol{\beta}} - \boldsymbol{\beta}^*)\|^2 = \sum_i \frac{\sigma^2}{n_S}\frac{t_i}{s_i}\left(1 - \frac{\lambda}{\sqrt{t_i}}\right)_+,$$

*where $\mathcal{B} = \{\boldsymbol{\beta}|\|\boldsymbol{\beta}\| \leq r\}$, and $\lambda = \lambda(r)$ is determined by $\frac{\sigma^2}{n_S}\sum_{i=1}^d \frac{1}{s_i}(\sqrt{t_i}/\lambda - 1)_+ = r^2$. The linear minimax estimator is given by:*

$$\hat{\boldsymbol{\beta}}_{MM} = \Sigma_T^{-1/2}U(I - \mathrm{diag}(\lambda/\sqrt{\boldsymbol{t}}))_+ U^\top \Sigma_T^{1/2}\hat{\boldsymbol{\beta}}_{SS}, \text{ where } \hat{\boldsymbol{\beta}}_{SS} = \frac{1}{n_S}\hat{\Sigma}_S^{-1}X_S^\top \boldsymbol{y}_S. \tag{2}$$

Since $r$ is unknown in practice, we could simply view either $r$ or directly $\lambda$ as the tuning parameter. We compare the functionality of $\lambda$ with that of ridge regression: $\hat{\boldsymbol{\beta}}_{\text{RR}}^\lambda = \arg\min_{\hat{\boldsymbol{\beta}}}\mathbb{E}\frac{1}{2n}\|X_S\hat{\boldsymbol{\beta}} - \boldsymbol{y}_S\|^2 + \frac{\lambda}{2}\|\hat{\boldsymbol{\beta}}\|^2 = (\hat{\Sigma}_S + \lambda I)^{-1}X_S^\top \boldsymbol{y}_S/n_S$. For both algorithms, $\lambda$ is to balance the bias and variance: $\lambda = 0$ gives an unbiased estimator, and a big $\lambda$ gives a (near) zero estimator with no variance. The difference is, our estimator shrinks some signal directions based on the value of $t_i$. The estimator tends to sacrifice the directions of signal where $t_i$ is smaller. Ridge regression, however, respects the value of $s_i$. A natural counterpart is for ridge to also regularize based on $\boldsymbol{t}$: let $\hat{\boldsymbol{\beta}}_{\text{RR},T}^\lambda = \arg\min \frac{1}{n}\|\Sigma_T^{1/2}(\boldsymbol{\beta} - \hat{\Sigma}_S^{-1}X_S^\top \boldsymbol{y}_S)\|^2 + \lambda\|\boldsymbol{\beta}\|^2 = (\Sigma_T + \lambda I)^{-1}\Sigma_T\hat{\boldsymbol{\beta}}_{\text{SS}}$. We will compare their performances in the experimental section.

---

[3]Throughout the paper $\hat{\Sigma}_S^{-1}$ could be replaced by pseudo-inverse and our algorithm also applies when $n < d$.

**Non-commutative covariance matrices.** For non-commutative covariate shift, we follow the same procedure. Our estimator is achieved by optimizing over $C$: $\hat{\boldsymbol{\beta}}_{\text{MM}} = C\hat{\boldsymbol{\beta}}_{\text{SS}}$:

$$R_L(\mathcal{B}) \equiv \min_{\hat{\boldsymbol{\beta}}=A\boldsymbol{y}_S} \max_{\boldsymbol{\beta}^*\in\mathcal{B}} \mathbb{E}\,\|\Sigma_T^{1/2}(\hat{\boldsymbol{\beta}}-\boldsymbol{\beta}^*)\|_2^2$$

$$= \min_{\hat{\boldsymbol{\beta}}=C\hat{\boldsymbol{\beta}}_{\text{SS}}} \max_{\|\boldsymbol{\beta}^*\|\le r} \left\{ \|\Sigma_T^{1/2}(C-I)\boldsymbol{\beta}^*\|_2^2 + \frac{\sigma^2}{n_S}\text{Tr}(\Sigma_T^{1/2}C\hat{\Sigma}_S^{-1}C^\top\Sigma_T^{1/2}) \right\} \qquad \text{(Claim 3.1)}$$

$$= \min_{\tau,C} \left\{ r^2\tau + \frac{\sigma^2}{n_S}\text{Tr}(\Sigma_T^{1/2}C\hat{\Sigma}_S^{-1}C^\top\Sigma_T^{1/2}) \right\},\ \text{s.t. } (C-I)^\top\Sigma_T(C-I) \preceq \tau I. \qquad (3)$$

Unlike the commutative case, this problem doesn't have a closed form solution, but is still solvable:

**Proposition 3.3.** *Problem* (3) *is a convex program and thus solvable.*

We achieve near-optimal minimax risk among all estimators under some conditions:

**Theorem 3.4** (Near minimaxity of linear estimators). *When $\Sigma_S, \Sigma_T$ commute, or $\Sigma_T$ is rank 1, the best linear estimator from* (2) *or* (3) *achieves near-optimal minimax risk: $L_\mathcal{B}(\hat{\boldsymbol{\beta}}_{MM}) = R_L(\mathcal{B}) \le 1.25R_N(\mathcal{B})$.*

Note that $R_N \le R_L$ by definition. Therefore 1) our estimator $\hat{\boldsymbol{\beta}}_{\text{MM}}$ is near-optimal, and 2) our lower bound for $R_N$ is tight. Lower bounds (without matching upper bounds) for general non-commutative problem is presented in (Kalan et al., 2020) and we improve their result for the commutative case and provide a matching algorithm. Their lower bound scales with $\frac{d}{n_S}\min_i\frac{t_i}{s_i}$ for large $r$, while ours becomes $\frac{1}{n_S}\sum_i\frac{t_i}{s_i}$. Our lower bound is always larger and thus tighter, and potentially arbitrarily larger when $\max_i\frac{t_i}{s_i}$ and $\min_i\frac{t_i}{s_i}$ are very different. We defer our proof to the appendix.

**Remark 3.1** (Benefit of minimax linear estimator). *Consider estimators from ridge regression: $\hat{\boldsymbol{\beta}}_{RR}^\lambda = \arg\min_{\hat{\boldsymbol{\beta}}} \mathbb{E}\frac{1}{2n}\|X_S\hat{\boldsymbol{\beta}} - \boldsymbol{y}_S\|^2 + \frac{\lambda}{2}\|\hat{\boldsymbol{\beta}}\|^2$. There is an example that $R_L(\mathcal{B}) \le \mathcal{O}(d^{-1/4}L_\mathcal{B}(\hat{\boldsymbol{\beta}}_{RR}^\lambda))$ even with the optimal hyperparameter $\lambda$.* [4]

**Remark 3.2** (Incorporating the randomness of source and target features). *For clean presentation purposes, in the main text we assume to have access to $\Sigma_T$. In practice, we will need to estimate $\Sigma_T$ by finite unlabeled samples from target domain. In Appendix C.1 we show that our estimator remains near-optimal if we have $\gg d$ unlabeled target samples under some standard light-tail assumptions.*
*Theorem 3.4 is comparing our estimator with the optimal nonlinear estimator using the same data $X_S$ from the source domain. In appendix C we compare our estimator with a stronger notion of linear estimator with infinite access to $p_S$ and show that our estimator is still within multiplicative factor of it.*

### 3.2 Linear minimax estimator with approximation error

Now we consider observations coming from nonlinear models: $\boldsymbol{y}_S = f^*(X_S) + \boldsymbol{z}$. Let $\boldsymbol{\beta}_S^* = \arg\min_{\boldsymbol{\beta}} \mathbb{E}_{\boldsymbol{x}\sim p_S, z\sim\mathcal{N}(0,\sigma^2)}[(f^*(\boldsymbol{x}) + z - \boldsymbol{\beta}^\top\boldsymbol{x})^2]$, and similarly for $\boldsymbol{\beta}_T^*$. Notice now even with $f^*$ unchanged across domains, the input distribution affects the best linear model. Approximation error is $a_S(\boldsymbol{x}) = f^*(\boldsymbol{x}) - \boldsymbol{x}^\top\boldsymbol{\beta}_S^*$ and vice versa for $a_T$.

---

[4] Note this goes without saying that our method can also be order-wise better than ordinary least square, which is a special case of ridge regression by setting $\lambda = 0$.

Define the reweighting vector $\boldsymbol{w} \in \mathbb{R}^n$ as $w_i = p_T(\boldsymbol{x}_i)/p_S(\boldsymbol{x}_i)$. We form unbiased estimator via

$$\hat{\boldsymbol{\beta}}_{\text{LS}} = \arg\min_{\boldsymbol{\beta}}\{\sum_i \frac{p_T(\boldsymbol{x}_i)}{p_S(\boldsymbol{x}_i)}(\boldsymbol{\beta}^\top \boldsymbol{x}_i - y_i)^2\} = (X_S^\top \operatorname{diag}(\boldsymbol{w})X_S)^{-1}(X_S^\top \operatorname{diag}(\boldsymbol{w})\boldsymbol{y}_S).$$

**Claim 3.5.** *$\hat{\boldsymbol{\beta}}_{LS}$ is asymptotically unbiased and normally distributed:*

$$\sqrt{n_S}(\hat{\boldsymbol{\beta}}_{LS} - \boldsymbol{\beta}_T^*) \xrightarrow{d} \mathcal{N}(0, \Sigma_T^{-1} \mathbb{E}_{\boldsymbol{x}\sim p_T}[p_T(\boldsymbol{x})/p_S(\boldsymbol{x})(a_T(\boldsymbol{x})^2 + \sigma^2)\boldsymbol{x}\boldsymbol{x}^\top]\Sigma_T^{-1}).$$

Denote by $m(\boldsymbol{x}) = a_T(\boldsymbol{x}) + z$. We want to minimize the worst case risk:

$$\min_{\hat{\boldsymbol{\beta}}=C\hat{\boldsymbol{\beta}}_{\text{LS}}} \max_{\boldsymbol{\beta}_T^* \in \mathcal{B}} \mathbb{E}\,\|\Sigma_T^{1/2}(\hat{\boldsymbol{\beta}} - \boldsymbol{\beta}_T^*)\|^2$$

$$\xrightarrow{d} \min_C \max_{\|\boldsymbol{\beta}_T^*\|\leq r} \left\{\|\Sigma_T^{1/2}(C-I)\boldsymbol{\beta}_T^*\|_2^2 + \frac{1}{n_S}\operatorname{Tr}(C\Sigma_T^{-1}\mathbb{E}_{p_T}[\frac{p_T(\boldsymbol{x})}{p_S(\boldsymbol{x})}m(\boldsymbol{x})^2\boldsymbol{x}\boldsymbol{x}^\top]\Sigma_T^{-1}C^\top\Sigma_T)\right\}$$

$$= \min_C \left\{\|(C-I)^\top\Sigma_T(C-I)\|_2 r^2 + \frac{1}{n_S}\operatorname{Tr}(C\Sigma_T^{-1}\mathbb{E}_{p_T}[\frac{p_T(\boldsymbol{x})}{p_S(\boldsymbol{x})}m(\boldsymbol{x})^2\boldsymbol{x}\boldsymbol{x}^\top]\Sigma_T^{-1}C^\top\Sigma_T)\right\}$$

Therefore our estimator is $\hat{\boldsymbol{\beta}}_{\text{MM}} \leftarrow \hat{C}\hat{\boldsymbol{\beta}}_{\text{LS}}$, where $\hat{C}$ finds

$$\hat{C} \leftarrow \arg\min_{\tau,C} \left\{r^2\tau + \frac{1}{n_S}\left\langle \frac{1}{n_S}\sum_i \frac{p_T^2(\boldsymbol{x})}{p_S^2(\boldsymbol{x})}(y_i - \boldsymbol{x}_i^\top\hat{\boldsymbol{\beta}}_{\text{LS}})^2\boldsymbol{x}_i\boldsymbol{x}_i^\top, \Sigma_T^{-1}C^\top\Sigma_T C\Sigma_T^{-1}\right\rangle\right\} \quad (4)$$

$$\text{s.t. } (C-I)^\top\Sigma_T(C-I) \preceq \tau I.$$

**Claim 3.6.** *Let $\mathcal{B} = \{\boldsymbol{\beta}|\|\boldsymbol{\beta}\| \leq r\}$, and $f^* \in \mathcal{F}$ is some compact symmetric function class: $f \in \mathcal{F} \Leftrightarrow -f \in \mathcal{F}$. Then linear minimax estimator is of the form $C\hat{\boldsymbol{\beta}}_{LS}$ for some $C$. When $\hat{C}$ solves Eqn. (4), $L_{\mathcal{B}}(\hat{\boldsymbol{\beta}}_{MM})$ asymptotically matches $R_L(\mathcal{B})$, the linear minimax risk.*

By reducing from $\boldsymbol{y}_S$ to $\hat{\boldsymbol{\beta}}_{\text{LS}}$ we eliminate $n - d$ dimensions, and this claim says that $X_S^\top\boldsymbol{y}_S$ is sufficient to predict $\boldsymbol{\beta}_T^*$. We note that $f^*$ is more general than a linear function and therefore the lower bound could only be larger than $R_N(\mathcal{B})$ defined in the previous section.

### 3.3 UTILIZE SOURCE AND TARGET LABELED DATA JOINTLY

In some scenarios we have moderate amount of labeled data from target domain as well. Then it is important to utilize the source and target labeled data jointly. Let $\boldsymbol{y}_S = X_S\boldsymbol{\beta}^* + \boldsymbol{z}_S$, $\boldsymbol{y}_T = X_T\boldsymbol{\beta}^* + \boldsymbol{z}_T$. We consider $X_S, X_T$ as deterministic variables, $\hat{\Sigma}_S^{-1}X_S^\top\boldsymbol{y}_S/n_S \sim \mathcal{N}(\boldsymbol{\beta}^*, \frac{\sigma^2}{n_S}\hat{\Sigma}_S^{-1})$ and $\hat{\Sigma}_T^{-1}X_T^\top\boldsymbol{y}_T/n_T \sim \mathcal{N}(\boldsymbol{\beta}^*, \frac{\sigma^2}{n_T}\hat{\Sigma}_T^{-1})$. Therefore conditioned on the observations $\boldsymbol{y}_S, \boldsymbol{y}_T$, a sufficient statistic for $\boldsymbol{\beta}^*$ is $\hat{\boldsymbol{\beta}}_{\text{SS}} := (n_S\hat{\Sigma}_S + n_T\hat{\Sigma}_T)^{-1}(X_S^\top\boldsymbol{y}_S + X_T^\top\boldsymbol{y}_T)$.

**Claim 3.7.** *$\hat{\boldsymbol{\beta}}_{SS}$ is an unbiased sufficient statistic of $\boldsymbol{\beta}^*$ with samples $\boldsymbol{y}_S, \boldsymbol{y}_T$. $\hat{\boldsymbol{\beta}}_{SS} \sim \mathcal{N}(\boldsymbol{\beta}^*, \sigma^2(n_S\hat{\Sigma}_S + n_T\hat{\Sigma}_T)^{-1})$.*

**Algorithm:** First consider the estimator $\hat{\boldsymbol{\beta}}_{\text{SS}} = (n_S\hat{\Sigma}_S + n_T\hat{\Sigma}_T)^{-1}(X_S^\top\boldsymbol{y}_S + X_T^\top\boldsymbol{y}_T)$. Next find the best linear function of $\hat{\boldsymbol{\beta}}_{\text{SS}}$:

$$\hat{\boldsymbol{\beta}}_{\text{MM}} = \arg\min_{C,\tau} r^2\tau + \sigma^2\operatorname{Tr}((n_S\hat{\Sigma}_S + n_T\hat{\Sigma}_T)^{-1}C^\top\Sigma_T C), \text{ s.t. } (C-I)^\top\Sigma_T(C-I) \preceq \tau.$$

**Proposition 3.8.** *The minimax estimator $\hat{\boldsymbol{\beta}}_{MM}$ is of the form $C\hat{\boldsymbol{\beta}}_{SS}$ for some $C$. When choosing $C$ with our proposed algorithm and when $\hat{\Sigma}_S$ commutes with $\hat{\Sigma}_T$ and $\Sigma_T$, we achieve the minimax risk $R_L(\mathcal{B}) \leq 1.25 R_N(\mathcal{B})$.*

## 4  NEAR MINIMAX ESTIMATOR WITH MODEL SHIFT

The general setting of transfer learning in linear regression involves both model shift and covariate shift. Namely, the generative model of the labels might be different: $\boldsymbol{y}_S = X_S \boldsymbol{\beta}_S^* + \boldsymbol{z}_S$, and $\boldsymbol{y}_T = X_T \boldsymbol{\beta}_T^* + \boldsymbol{z}_T$. Denote by $\boldsymbol{\delta} := \boldsymbol{\beta}_S^* - \boldsymbol{\beta}_T^*$ as the model shift. We are interested in the minimax linear estimator when $\|\boldsymbol{\delta}\| \leq \gamma$ and $\|\boldsymbol{\beta}_T^*\| \leq r$. Thus our problem becomes to find minimax estimator for $\boldsymbol{\beta}_T^* \in \mathcal{B} = \{\boldsymbol{\beta} | \|\boldsymbol{\beta}\| \leq r\}$ from $\boldsymbol{y}_S, \boldsymbol{y}_T$.

**Algorithm:**  First consider a sufficient statistic $(\bar{\boldsymbol{\beta}}_S, \bar{\boldsymbol{\beta}}_T)$ for $(\boldsymbol{\beta}_T^*, \boldsymbol{\delta})$. Here $\bar{\boldsymbol{\beta}}_S = \hat{\Sigma}_S^{-1} X_S^\top \boldsymbol{y}_S / n_S \sim \mathcal{N}(\boldsymbol{\beta}_T^* + \boldsymbol{\delta}, \frac{\sigma^2}{n_S} \hat{\Sigma}_S^{-1})$, and $\bar{\boldsymbol{\beta}}_T = \hat{\Sigma}_T^{-1} X_T^\top \boldsymbol{y}_T / n_T \sim \mathcal{N}(\boldsymbol{\beta}_T^*, \frac{\sigma^2}{n_T} \hat{\Sigma}_T^{-1})$. Then consider the best linear estimator on top of it: $\hat{\boldsymbol{\beta}} = A_1 \bar{\boldsymbol{\beta}}_S + A_2 \bar{\boldsymbol{\beta}}_T$. Write $\Delta = \{\boldsymbol{\delta} | \|\boldsymbol{\delta}\| \leq \gamma\}$ and $L_{\mathcal{B},\Delta}(\hat{\boldsymbol{\beta}}) := \max_{\boldsymbol{\beta}_T^* \in \mathcal{B}, \boldsymbol{\delta} \in \Delta} \|\Sigma_T^{1/2}(\hat{\boldsymbol{\beta}} - \boldsymbol{\beta}_T^*)\|^2$.

$$
\begin{aligned}
R_L(\mathcal{B}, \Delta) &:= \min_{\hat{\boldsymbol{\beta}} = A_1 \bar{\boldsymbol{\beta}}_S + A_2 \bar{\boldsymbol{\beta}}_T} L_{\mathcal{B},\Delta}(\hat{\boldsymbol{\beta}}) \\
&\leq \min_{A_1, A_2} \max_{\|\boldsymbol{\beta}_T^*\| \leq r, \|\boldsymbol{\delta}\| \leq \gamma} \Big\{ 2\|\Sigma_T^{1/2}((A_1 + A_2 - I)\boldsymbol{\beta}_T^*\|^2 + 2\|\Sigma_T^{1/2} A_1 \boldsymbol{\delta}\|^2 \qquad (5) \\
&\qquad + \frac{\sigma^2}{n_S}\mathrm{Tr}(A_1 \hat{\Sigma}_S^{-1} A_1^\top) + \frac{\sigma^2}{n_T}\mathrm{Tr}(A_2 \hat{\Sigma}_T^{-1} A_2^\top) \Big\} \qquad \text{(AM-GM)} \\
&= \min_{A_1, A_2} \Big\{ 2\|\Sigma_T^{1/2}((A_1 + A_2 - I)\|_2^2 r^2 + 2\|\Sigma_T^{1/2} A_1\|_2^2 \gamma^2 \\
&\qquad + \frac{\sigma^2}{n_S}\mathrm{Tr}(A_1 \hat{\Sigma}_S^{-1} A_1^\top) + \frac{\sigma^2}{n_T}\mathrm{Tr}(A_2 \hat{\Sigma}_T^{-1} A_2^\top) =: r_{\mathcal{B},\Delta}(A_1, A_2) \Big\}.
\end{aligned}
$$

Therefore we optimize over this upper bound and reformulate the problem as a convex program:

$$
\begin{aligned}
(\hat{A}_1, \hat{A}_2) &\leftarrow \arg\min_{A_1, A_2, a, b} \quad \Big\{ 2ar^2 + 2b\gamma^2 + \frac{\sigma^2}{n_S}\mathrm{Tr}(A_1 \hat{\Sigma}_S^{-1} A_1^\top) + \frac{\sigma^2}{n_T}\mathrm{Tr}(A_2 \hat{\Sigma}_T^{-1} A_2^\top) \Big\} \\
&\text{s.t.} \qquad (A_1 + A_2 - I)^\top \Sigma_T (A_1 + A_2 - I) \preceq aI, \ A_1^\top \Sigma_T A_1 \preceq bI. \quad (6)
\end{aligned}
$$

Our estimator is given by: $\hat{\boldsymbol{\beta}}_{\mathrm{MM}} = \hat{A}_1 \bar{\boldsymbol{\beta}}_S + \hat{A}_2 \bar{\boldsymbol{\beta}}_T$. Since $\hat{\boldsymbol{\beta}}_{\mathrm{MM}}$ is a relaxation of the linear minimax estimator, it is important to understand how well $\hat{\boldsymbol{\beta}}_{\mathrm{MM}}$ performs on the original objective:

**Claim 4.1.** $R_L(\mathcal{B}, \Delta) \leq L_{\mathcal{B},\Delta}(\hat{\boldsymbol{\beta}}_{MM}) \leq 2R_L(\mathcal{B}, \Delta)$.

Finally we show with the relaxation we still achieve a near-optimal estimator even among all nonlinear rules.

**Theorem 4.2.** *When $\Sigma_T$ commutes with $\hat{\Sigma}_S$, it satisfies:*

$$
L_{\mathcal{B},\Delta}(\hat{\boldsymbol{\beta}}_{MM}) := \max_{\boldsymbol{\beta}_T^* \in \mathcal{B}, \boldsymbol{\delta} \in \Delta} \|\Sigma_T^{1/2}(\hat{\boldsymbol{\beta}}_{MM} - \boldsymbol{\beta}_T^*)\|^2 \leq 27 R_N(\mathcal{B}, \Delta).
$$

*Here $R_N(\mathcal{B}, \Delta) := \min_{\hat{\boldsymbol{\beta}}(\boldsymbol{y}_S, \boldsymbol{y}_T)} \max_{\boldsymbol{\beta}_T^* \in \mathcal{B}, \boldsymbol{\delta} \in \Delta} \|\Sigma_T^{1/2}(\hat{\boldsymbol{\beta}} - \boldsymbol{\beta}_T^*)\|$ is the minimax risk.*

*Proof sketch of Theorem 4.2.* For the ease of understanding, we provide a simple proof sketch when $\Sigma_S = \Sigma_T$ are diagonal. We first define the hardest hyperrectangular subproblem. Let $\mathcal{B}(\boldsymbol{\tau}) = \{\boldsymbol{b} : |\beta_i| \leq \tau_i\}$ be a subset of $\mathcal{B}$ and similarly for $\Delta(\boldsymbol{\zeta})$. We show that $R_L(\mathcal{B}, \Delta) = $

$\max_{\boldsymbol{\tau}\in\mathcal{B},\boldsymbol{\zeta}\in\Delta} R_L(\mathcal{B}(\boldsymbol{\tau}),\Delta(\boldsymbol{\zeta}))$, and clearly $R_N(\mathcal{B},\Delta) \geq \max_{\boldsymbol{\tau}\in\mathcal{B},\boldsymbol{\zeta}\in\Delta} R_N(\mathcal{B}(\boldsymbol{\tau}),\Delta(\boldsymbol{\zeta}))$. Meanwhile we show when the sets are hyperrectangles the minimax (linear) risk could be decomposed to 1-d problems: $R_L(\mathcal{B}(\boldsymbol{\tau}),\Delta(\boldsymbol{\zeta})) = \sum_i R_L(\tau_i,\zeta_i)$. Each $R_L(\tau_i,\zeta_i)$ is the linear minimax risk to estimate $\beta_i$ from $x \sim \mathcal{N}(\beta_i + \delta_i, 1)$ and $y \sim \mathcal{N}(\beta_i, 1)$ where $|\beta_i| \leq \tau_i$ and $|\delta_i| \leq \zeta_i$. This 1-d problem for linear risk has a closed form solution, and the minimax risk can be lower bounded using Le Cam's two point lemma. We show $R_L(\tau_i,\zeta_i) \leq 13.5 R_N(\tau_i,\zeta_i)$ and therefore:

$$\frac{1}{2}L_{\mathcal{B},\Delta}(\hat{\boldsymbol{\beta}}_{\text{MM}}) \overset{\text{Claim 4.1}}{\leq} R_L(\mathcal{B},\Delta) \overset{\text{Lemma } B.2}{=} \max_{\boldsymbol{\tau}\in\mathcal{B},\boldsymbol{\zeta}\in\Delta} R_L(\mathcal{B}(\boldsymbol{\tau}),\Delta(\boldsymbol{\zeta}))$$

$$\overset{\text{Prop } B.4.a}{=} \max_{\boldsymbol{\tau}\in\mathcal{B},\boldsymbol{\zeta}\in\Delta} \sum_i R_L(\tau_i,\zeta_i) \overset{\text{Lemma } B.6}{\leq} \max_{\boldsymbol{\tau}\in\mathcal{B},\boldsymbol{\zeta}\in\Delta} 13.5\sum_i R_N(\tau_i,\zeta_i)$$

$$\overset{\text{Prop } B.4.b}{=} 13.5 \max_{\boldsymbol{\tau}\in\mathcal{B},\boldsymbol{\zeta}\in\Delta} R_N(\mathcal{B}(\boldsymbol{\tau}),\Delta(\boldsymbol{\zeta})) \leq 13.5 R_N(\mathcal{B},\Delta).$$

$\square$

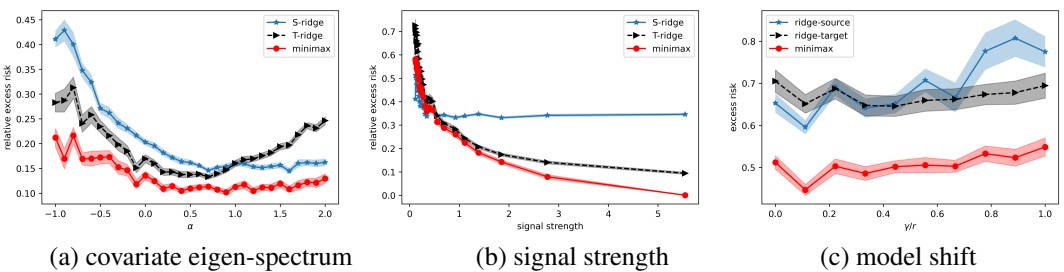

(a) covariate eigen-spectrum     (b) signal strength     (c) model shift

Figure 1: *Performance comparisons.* (a): The x-axis $\alpha$ defines the spread of eigen-spectrum of $\Sigma_S$: $s_i \propto 1/i^{\alpha}, t_i \propto 1/i$. (b) x-axis is the normalized value of signal strength: $\|\Sigma_T \boldsymbol{\beta}^*\|/r$. (c) X-axis is the model shift measured by $\gamma/r$. Performance with standard error bar is from 40 runs.

## 5   EXPERIMENTS

Our estimators are provably near optimal for the worst case $\boldsymbol{\beta}^*$. However, it remains unknown whether on average they outperform other baselines. With synthetic data we explore the performances with random $\boldsymbol{\beta}^*$. We are also interested to investigate the conditions when we win more.

**Setup.** We set $n_S = 2000, d = 50, \sigma = 1, r = \sqrt{d}$. For each setting, we sample $\boldsymbol{\beta}_T^*$ from standard normal distribution and rescale it to be norm $r$. We assume to know $\Sigma_T$. We compare our estimator with ridge regression (S-ridge) and a variant of ridge regression transformed to target domain (T-ridge): $\hat{\boldsymbol{\beta}}_{\text{RR},T}^{\lambda} = \arg\min \frac{1}{n}\|\Sigma_T^{1/2}(\boldsymbol{\beta} - \hat{\Sigma}_S^{-1} X_S^{\top} \boldsymbol{y}_S)\|^2 + \lambda\|\boldsymbol{\beta}\|^2 = (\Sigma_T + \lambda I)^{-1}\Sigma_T \hat{\boldsymbol{\beta}}_{\text{SS}}$.

**Covariate shift.** In order to understand the effect of covariate shift on our algorithm, we consider three types of settings, each with a unique varying factor that influences the performance: 1) covariate eigenvalue shift with shared eigenspace; 2) covariate eigenspace shift with fixed eigenvalues[5];

---

[5]We leave this result in appendix since performance appears invariant to this factor.

3) signal strength change. We also have an additional 200 labeled data from target domain as validation set only for hyper-parameter tuning.

**Model shift.** Next we consider the problem with model shift. We sample a random $\boldsymbol{\delta}$ with norm $\gamma$ varying from 0 to $r = \sqrt{d}$ and observe data generated by $\boldsymbol{y}_S = X_S(\boldsymbol{\beta}_T^* + \boldsymbol{\delta}) + \boldsymbol{z}_S \in \mathbb{R}^{2000}, \boldsymbol{z}_S \sim \mathcal{N}(0, I)$ and $\boldsymbol{y}_T = X_T\boldsymbol{\beta}_T^* + \boldsymbol{z}_T \in \mathbb{R}^{500}, \boldsymbol{z}_T \sim \mathcal{N}(0, I)$. We compare our estimator with two baselines: "ridge-source" denotes ridge regression using only source data, and "ridge-target" is from ridge regression with target data.

Figure 1 demonstrates the better performance of our estimator in all circumstances. From (a) we see that with more discrepancy between $\Sigma_S$ and $\Sigma_T$, our estimator tends to perform better. (b) shows our estimator is better when the signal is relatively stronger. From (c) we can see that with the increasing model shift measured by $\gamma/r$, ridge-source becomes worse and is outperformed by ridge-target that remains unchanged. Our estimator becomes slightly worse as well due to the less utility from source data, but remains the best among others. When $\gamma/r \approx 0.2$, our method has the most improvement in percentage compared to the best result among ridge-source and ridge-target.

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

## A    OMITTED PROOF FOR MINIMAX ESTIMATOR WITH COVARIATE SHIFT

### A.1    PINSKER'S THEOREM AND COVARIATE SHIFT WITH LINEAR MODEL

**Theorem A.1** (Pinsker's Theorem). *Suppose the obervations follow sequence model $y_i = \theta_i^* + \epsilon_i z_i, \epsilon_i > 0, i \in [d]$, and $\Theta$ is an ellipsoid in $\mathbb{R}^d$: $\Theta = \Theta(a, C) = \{\theta : \sum_i a_i^2 \theta_i^2 \le C^2\}$. Then the minimax linear risk*

$$R_L(\Theta) := \min_{\hat{\boldsymbol{\theta}} \text{ linear}} \max_{\boldsymbol{\theta}^* \in \Theta} \mathbb{E} \, \|\hat{\boldsymbol{\theta}}(\boldsymbol{y}) - \boldsymbol{\theta}^*\|^2$$

$$= \sum_i \epsilon_i^2 (1 - a_i/\mu)_+,$$

*where $\mu = \mu(C)$ is determined by*

$$\sum_{i=1}^{d} \epsilon_i^2 a_i (\mu - a_i)_+ = C^2.$$

*The linear minimax estimator is given by*

$$\hat{\theta}_i^*(y) = c_i^* y_i = (1 - a_i/\mu)_+ y_i, \tag{7}$$

*and is Bayes for a Gaussian prior $\pi_C$ having independent components $\theta_i \sim \mathcal{N}(0, \tau_i^2)$ with $\tau_i^* = \epsilon_i^2 (\mu/a_i - 1)_+$.*

Our theorem 3.2 is to connect our parameter $\boldsymbol{\beta}^*$ to the $\boldsymbol{\theta}^*$ in pinsker's theorem. First we show that reformulating the problem from a linear map of $n$ dimensional observations $\boldsymbol{y}_S$ to a linear map on the $d$-dimensional statistic $\hat{\boldsymbol{\beta}}_{SS}$ is sufficient, i.e., Claim 3.1:

*Proof of Claim 3.1.* This is to show that if $\hat{\boldsymbol{\beta}}(\boldsymbol{y}_S) := A\boldsymbol{y}_S$ is a minimax linear estimator, each row vector of $A \in \mathbb{R}^{d \times n}$ is in the column span of $X_S$. Write $A = A_1 X_S^\top + A_2 W^\top$ where $W \in \mathbb{R}^{n \times (n-d)}$, columns of which forms the orthonormal complement for the column space of $X_S$. Equivalently we want to show $A_2 = 0$. We have

$$\begin{aligned}
R_L(\mathcal{B}) &\equiv \min_{\hat{\boldsymbol{\beta}} = A\boldsymbol{y}} \max_{\boldsymbol{\beta}^* \in \mathcal{B}} \mathbb{E} \, \|\Sigma_T^{1/2}(\hat{\boldsymbol{\beta}} - \boldsymbol{\beta}^*)\|^2 \\
&= \min_{A_1, A_2} \max_{\boldsymbol{\beta}^* \in \mathcal{B}} \mathbb{E} \, \|\Sigma_T^{1/2}((A_1 X_S^\top + A_2 W^\top)\boldsymbol{y}_S - \boldsymbol{\beta}^*)\|^2 \\
&= \min_{A_1, A_2} \max_{\boldsymbol{\beta}^* \in \mathcal{B}} \mathbb{E} \, \|\Sigma_T^{1/2}(A_1 X_S^\top (X_S \boldsymbol{\beta}^* + \boldsymbol{z}) + A_2 W^\top \boldsymbol{z} - \boldsymbol{\beta}^*)\|^2 \quad \text{(Since } W^\top X_S = 0\text{)} \\
&= \min_{A_1, A_2} \max_{\boldsymbol{\beta}^* \in \mathcal{B}} \Big\{ \|\Sigma_T^{1/2}(A_1 X_S^\top X_S - I)\boldsymbol{\beta}^*\|^2 + \mathbb{E} \, \|\Sigma_T^{1/2} A_1 X_S^\top \boldsymbol{z}\|^2 \\
&\qquad + \mathbb{E} \, \|\Sigma_T^{1/2} A_2 W^\top \boldsymbol{z}\|^2 + \mathbb{E} \left\langle \Sigma_T^{1/2} A_1 X_S^\top \boldsymbol{z}, \Sigma_T^{1/2} A_2 W^\top \boldsymbol{z} \right\rangle \Big\} \\
&\qquad\qquad\qquad\qquad\qquad\qquad \text{(Other cross terms vanish since } \mathbb{E}[\boldsymbol{z}] = \boldsymbol{0}\text{)} \\
&= \min_{A_1, A_2} \max_{\boldsymbol{\beta}^* \in \mathcal{B}} \Big\{ \|\Sigma_T^{1/2}(A_1 X_S^\top X_S - I)\boldsymbol{\beta}^*\|^2 + \mathbb{E} \, \|\Sigma_T^{1/2} A_1 X_S^\top \boldsymbol{z}\|^2 + \mathbb{E} \, \|\Sigma_T^{1/2} A_2 W^\top \boldsymbol{z}\|^2, \Big\}
\end{aligned}$$

where the last equation is because

$$\mathbb{E} \left\langle \Sigma_T^{1/2} A_1 X_S^\top \boldsymbol{z}, \Sigma_T^{1/2} A_2 W^\top \boldsymbol{z} \right\rangle = \mathbb{E} \left[ \operatorname{Tr} \left[ \Sigma_T^{1/2} A_1 X_S^\top \boldsymbol{z} \boldsymbol{z} W A_2^\top \Sigma_T \right] \right]$$

$$= \mathrm{Tr}\left[\Sigma_T^{1/2} A_1 X_S^\top \, \mathbb{E}[\boldsymbol{z}\boldsymbol{z}^\top] W A_2^\top \Sigma_T\right] = \sigma^2 \mathrm{Tr}\left[\Sigma_T^{1/2} A_1 X_S^\top W A_2^\top \Sigma_T\right] = 0.$$

Clearly, at min-max point, without loss of generality we can take $A_2 = 0$. □

Formally the proof for Theorem 3.2 is presented here:

*Proof of Theorem 3.2.* To use Pinsker's theorem to prove Theorem 3.2, we simply need to transform the problem match its setting. Let $\boldsymbol{y}_T = \Sigma_T^{1/2} \hat{\Sigma}_S^{-1} X_S^\top \boldsymbol{y}_S / n_S = \boldsymbol{\theta}_T^* + \boldsymbol{z}_T$, where $\boldsymbol{\theta}_T^* = U^\top \Sigma_T^{1/2} \boldsymbol{\beta}^*$ and $\boldsymbol{z}_T \sim \mathcal{N}(0, \sigma^2 \mathrm{diag}([t_i/s_i]_{i=1}^d)/n_S)$. The set for $\theta_T^*$ is $\Theta = \{\boldsymbol{\theta} | \|\Sigma_T^{-1/2} U \boldsymbol{\theta}\| \leq r\}$, i.e., $\Theta = \{\theta | \sum_i \boldsymbol{\theta}_i^2 / t_i \leq r^2\}$.

Now with Pinsker's theorem, $\hat{\boldsymbol{\theta}}(\boldsymbol{y}_T)_i = (1 - 1/(\mu\sqrt{t_i}))_+ (y_T)_i$ is the best linear estimator for $\boldsymbol{\theta}_T^*$, where $\mu = \mu(r)$ solves

$$\frac{\sigma^2}{n_S} \sum_{i=1}^d \frac{\sqrt{t_i}}{s_i}(\mu - \frac{1}{\sqrt{t_i}})_+ = r^2. \tag{8}$$

Connecting to the original problem, we get that the best estimator for $\Sigma_T^{1/2}\boldsymbol{\beta}^*$ is $U(I - \frac{1}{\mu}\mathrm{diag}([1/\sqrt{t_i}]_{i=1}^d))\boldsymbol{y}_T = U(I - \frac{1}{\mu}\mathrm{diag}([1/\sqrt{t_i}]_{i=1}^d))U^\top \Sigma_T^{1/2}\Sigma_S^{-1}X_S^\top \boldsymbol{y}_S/n_S$.

□

## A.2 Omitted proof for noncommute covariance matrices

**Convex program.** Our estimator for $\boldsymbol{\beta}^*$ can be achieved through convex programming:

*Proof of Proposition 3.3.* First note the objective function is quadratic in $C$ and linear in $\tau$, therefore we only need to prove the constraint $S = \{(C, \tau) | (C - I)^\top \Sigma_T (C - I) \preceq \tau I\}$ is a convex set. Notice for $(C_1, \tau_1), (C_2, \tau_2) \in S$, i.e., $(C_i - I)^\top \Sigma_T (C_i - I) \preceq \tau_i I, i \in \{1, 2\}$. We simply need to prove for $C_\alpha := \alpha C_1 + (1 - \alpha)C_2, \tau_\alpha := \tau_1\alpha + \tau_2(1 - \alpha), (C_\alpha - I)^\top \Sigma_T (C_\alpha - I) \preceq \tau_\alpha I$ for any $\alpha \in [0, 1]$. First, notice $(C_1 - C_2)^\top \Sigma_T (C_1 - C_2) \succeq 0$. Next,

$$(C_\alpha - I)^\top \Sigma_T (C_\alpha - I)$$
$$= \alpha(C_1 - I)^\top \Sigma_T (C_1 - I) + (1 - \alpha)(C_2 - I)^\top \Sigma_T (C_2 - I)$$
$$\quad - \alpha(1 - \alpha)(C_1 - C_2)^\top \Sigma_T (C_1 - C_2)$$
$$\preceq \alpha(C_1 - I)^\top \Sigma_T (C_1 - I) + (1 - \alpha)(C_2 - I)^\top \Sigma_T (C_2 - I)$$
$$\preceq \tau_\alpha I.$$

□

**Benefit of our estimator.** Compared to ridge regression, our estimator could possibly achieve much better $(d^{-1/4})$ improvements:

*Proof of Remark 3.1.* We consider diagonal covariance matrices $\hat{\Sigma}_S = \mathrm{diag}(\boldsymbol{s}), \Sigma_T = \mathrm{diag}(\boldsymbol{t})$, $\sigma = 1$. First we calculate the expected risk obtained with ridge regression: $\hat{\boldsymbol{\beta}}_{RR}^\lambda = (X_S^\top X_S / n + $

$$\lambda I)^{-1} X_S^\top \boldsymbol{y}_S / n_S \sim \mathcal{N}((\hat{\Sigma}_S + \lambda I)^{-1} \Sigma_S \boldsymbol{\beta}^*, 1/n_S (\Sigma_S + \lambda I)^{-2} \Sigma_S).$$

$$
\begin{aligned}
L_{\mathcal{B}}(\boldsymbol{\beta}_{\mathrm{RR}}^\lambda) &= \max_{\boldsymbol{\beta}^* \in \mathcal{B}} \mathbb{E}_{\boldsymbol{y}_S} \| \Sigma_T^{1/2} (\hat{\boldsymbol{\beta}}_{\mathrm{RR}}^\lambda(\boldsymbol{y}_S) - \boldsymbol{\beta}^*) \|^2 \\
&= \max_{\boldsymbol{\beta}^* \in \mathcal{B}} \| \Sigma_T^{1/2} ((\hat{\Sigma}_S + \lambda I)^{-1} \hat{\Sigma}_S - I) \boldsymbol{\beta}^* \|^2 + \mathrm{Tr}(\frac{1}{n_S}(\hat{\Sigma}_S + \lambda I)^{-2} \hat{\Sigma}_S \Sigma_T) \\
&= \max_i r^2 \left( \frac{\sqrt{t_i} s_i}{s_i + \lambda} - \sqrt{t_i} \right)^2 + \sum_i \frac{1}{n_S} \frac{t_i s_i}{(s_i + \lambda)^2}.
\end{aligned}
$$

Compared to our risk:

$$R_L(\mathcal{B}) = \sum_i \frac{1}{n_S} \frac{t_i}{s_i} (1 - \frac{1}{\sqrt{t_i}\mu})_+,$$

where $\frac{1}{n} \sum_{i=1}^d \frac{\sqrt{t_i}}{s_i} (\mu - \frac{1}{\sqrt{t_i}})_+ = r^2$. Let $r^2 = \frac{\sqrt{d}}{n_S}$, $s_i = 1, \forall i, t_i = 1, \forall i \in [d_0], t_i = d^{-1/2}, d_0 < i \le d$, where $d_0 = \frac{\sqrt{d}}{d^{1/4}-1} \approx d^{1/4}$. Then $\mu = 1$, and $R_L(\mathcal{B}) = \frac{d^{1/4}}{n}$. In this case,

$$
\begin{aligned}
&\min_\lambda \max_i r^2 \left( \frac{\sqrt{t_i} s_i}{s_i + \lambda} - \sqrt{t_i} \right)^2 + \sum_i \frac{1}{n_S} \frac{t_i s_i}{(s_i + \lambda)^2} \\
&= \min_\lambda \max_i \frac{\sqrt{d}}{n} \left( \frac{\sqrt{t_i}}{1 + \lambda} - \sqrt{t_i} \right)^2 + \sum_i \frac{1}{n_S} \frac{t_i}{(1 + \lambda)^2} \ge \min_\lambda \frac{\sqrt{d}}{n} \frac{\lambda^2}{(1 + \lambda)^2} + \frac{\sqrt{d}}{n} \frac{1}{(1 + \lambda)^2} \\
&\ge \frac{\sqrt{d}}{2n}.
\end{aligned}
$$

Therefore $\min_\lambda L_{\mathcal{B}}(\hat{\boldsymbol{\beta}}_{\mathrm{RR}}^\lambda) \ge d^{1/4} R_L(\mathcal{B})/2$. $\qquad\square$

**Near minimax risk.** Even among all nonlinear estimators, our estimator is within 1.25 of the minimax risk:

*Proof of Theorem 3.4.* First we note that for both linear and nonlinear estimators, it is sufficient to use $\hat{\boldsymbol{\beta}}_{\mathrm{SS}}$ instead of the original observations $\boldsymbol{y}_S$. See Lemma A.2 and its corollary. Therefore it suffices to do the following reformulations of the problem.

When $\Sigma_S$ and $\Sigma_T$ commute, we formulate the problem as the following Gaussian sequence model. Recall $\hat{\Sigma}_S = U \mathrm{diag}(\boldsymbol{s}) U^\top, \Sigma_T = U \mathrm{diag}(\boldsymbol{t}) U^\top$. Let $\boldsymbol{\theta}^* = U^\top \Sigma_T^{1/2} \boldsymbol{\beta}^*$, and $\boldsymbol{y} = U^\top \Sigma_T^{1/2} \hat{\boldsymbol{\beta}}_{\mathrm{SS}} \sim \mathcal{N}(\boldsymbol{\theta}^*, \frac{\sigma^2}{n_S} \mathrm{diag}(\boldsymbol{t}/\boldsymbol{s}))$. Our objective of minimizing $\| \Sigma_T^{1/2}(\hat{\boldsymbol{\beta}}(\boldsymbol{y}_S) - \hat{\boldsymbol{\beta}}^*) \|$ from linear estimator is equivalent to minimizing $\| U(\hat{\boldsymbol{\theta}}(\boldsymbol{y}) - \hat{\boldsymbol{\theta}}^*) \| = \| \hat{\boldsymbol{\theta}}(\boldsymbol{y}) - \hat{\boldsymbol{\theta}}^* \|$ from linear estimator.

The set for the parameter that satisfies $\boldsymbol{\theta}^* = U^\top \Sigma_T^{1/2} \boldsymbol{\beta}^*, \| \boldsymbol{\beta}^* \| \le r$ is equivalent to $\| \Sigma_T^{-1/2} U \boldsymbol{\theta}^* \| \le r \Leftrightarrow \| \theta_i^* / \sqrt{t_i} \| \le r$ is an axis-aligned ellipsoid. Then we could directly derive our result from Corollary 4.26 from Johnstone (2011). Note that this result is a special case of Theorem 4.2 and we have provided a detailed proof in Section B. Therefore here we save further descriptions.

For the case when $\Sigma_T = \boldsymbol{a}\boldsymbol{a}^\top$ is rank-1, the objective function becomes:

$$R_L^*(\mathcal{B}) = \min_{\boldsymbol{\beta}^* \text{ linear}} \max_{\boldsymbol{\beta} \in \mathcal{B}} \mathbb{E}(\boldsymbol{a}^\top (\hat{\boldsymbol{\beta}}(\boldsymbol{y}_S) - \boldsymbol{\beta}^*))^2.$$

Then the result could be derived from Corollary 1 of Donoho (1994), which reformulate the problem to the hardest one-dimensional problem which becomes tractable.

$\square$

In the proof above, we equate the best nonlinear estimator on $\boldsymbol{y}_S$ as the best nonlinear estimator on $\hat{\beta}_{\mathrm{SS}}$. The reasoning is as follows:

**Lemma A.2** (Sufficient statistic is enough to achieve a best estimator). *Consider the statistical problem of estimating $\boldsymbol{\beta}^* \in \mathcal{B}$ from observations $\boldsymbol{y} \in \mathcal{Y}$. $\mathcal{B}$ $\ell^2$-compact. If $S(\boldsymbol{y})$ is a sufficient statistic of $\boldsymbol{\beta}^*$, then the best estimator that achieves $\min_{\hat{\beta}} \max_{\mathcal{B}} \ell(\hat{\boldsymbol{\beta}}, \boldsymbol{\beta}^*)$ is of the form $\hat{\boldsymbol{\beta}} = f(S(\boldsymbol{y}))$ with some function $f$, for any loss $\ell : \mathcal{Y} \rightarrow [0, \infty)$.*

This Lemma is restated from Proposition 3.13 from Johnstone (2011).

**Corollary A.3** (Corollary of Lemma A.2). *Under the same setting of Lemma A.2, $R_N(\mathcal{B})$ is achieved with the form $\hat{\boldsymbol{\beta}} = f(S(\boldsymbol{y}))$.*

### A.3 OMITTED PROOF WITH APPROXIMATION ERROR

**Unbiased estimator for $\hat{\beta}_T^*$.**

*Proof of Claim 3.5.*

$$
\begin{aligned}
\hat{\boldsymbol{\beta}}_{\mathrm{LS}} - \boldsymbol{\beta}_T^* &= (X_S^\top \operatorname{diag}(\boldsymbol{w}) X_S)^{-1} (X_S^\top \operatorname{diag}(\boldsymbol{w}) \boldsymbol{y}) - \boldsymbol{\beta}_T^* \\
&= (X_S^\top \operatorname{diag}(\boldsymbol{w}) X_S)^{-1} (X_S^\top \operatorname{diag}(\boldsymbol{w}) (X_S \boldsymbol{\beta}_T^* + \boldsymbol{a}_T + \boldsymbol{z})) - \boldsymbol{\beta}_T^* \\
&= (X_S^\top \operatorname{diag}(\boldsymbol{w}) X_S)^{-1} (X_S^\top \operatorname{diag}(\boldsymbol{w}) (\boldsymbol{a}_T + \boldsymbol{z}))
\end{aligned}
$$

Notice $\mathbb{E}_{\boldsymbol{x} \sim p_S}[\boldsymbol{x} a_T(\boldsymbol{x}) \frac{p_T(\boldsymbol{x})}{p_S(\boldsymbol{x})}] = \mathbb{E}_{\boldsymbol{x} \sim p_T}[\boldsymbol{x} a_T(\boldsymbol{x})] = 0$. This is due to the KKT condition for the minimizer of $l(\boldsymbol{\beta}) := \mathbb{E}_{\boldsymbol{x} \sim p_T} \|f^*(\boldsymbol{x}) - \boldsymbol{\beta}^\top \boldsymbol{x}\|^2$ at $\boldsymbol{\beta}_T^*$: $\nabla_{\boldsymbol{\beta}} f(\boldsymbol{\beta}^*) = 0 \rightarrow \mathbb{E}_{\boldsymbol{x} \sim p_T}[\boldsymbol{x}(f^* - \boldsymbol{x}^\top \boldsymbol{\beta}_T^*)] = 0$, i.e., $\mathbb{E}_{\boldsymbol{x} \sim p_T}[\boldsymbol{x} a_T(\boldsymbol{x})] = 0$. Next we have: $\mathbb{E}_{\boldsymbol{x}_i \sim p_S}[X_S^\top \operatorname{diag}(\boldsymbol{w}) X_S] = \mathbb{E}_{\boldsymbol{x}_i \sim p_S} \sum_{i=1}^n \frac{p_T(\boldsymbol{x}_i)}{p_S(\boldsymbol{x}_i)} \boldsymbol{x}_i \boldsymbol{x}_i^\top = \mathbb{E}_{\boldsymbol{x}_j \sim p_T} \sum_{j=1}^n [\boldsymbol{x}_j \boldsymbol{x}_j^\top] = n_S \Sigma_T$. Therefore

$$
\hat{\boldsymbol{\beta}}_{\mathrm{LS}} - \boldsymbol{\beta}_T^* \rightarrow \mathcal{N}(0, \frac{1}{n_S} \Sigma_T^{-1} \mathbb{E}_{\boldsymbol{x} \sim p_T}[p_T(\boldsymbol{x})/p_S(\boldsymbol{x})(a_T(\boldsymbol{x})^2 + \sigma^2) \boldsymbol{x} \boldsymbol{x}^\top] \Sigma_T^{-1}).
$$

$\square$

*Proof of Claim 3.6.* Recall $X_S = [\boldsymbol{x}_1^\top | \boldsymbol{x}_2^\top | \cdots | \boldsymbol{x}_n^\top]^\top \in \mathbb{R}^{n \times d}$, with $\boldsymbol{x}_i, \forall i \in [n]$ drawn from $p_S$, and $\boldsymbol{a}_T = [a_T(\boldsymbol{x}_1), a_T(\boldsymbol{x}_2), \cdots a_T(\boldsymbol{x}_n)]^\top \in \mathbb{R}^n$, $\boldsymbol{y} = [y(\boldsymbol{x}_1), y(\boldsymbol{x}_2), \cdots, y(\boldsymbol{x}_n)]^\top \in \mathbb{R}^n$, noise $\boldsymbol{z} = \boldsymbol{y} - f^*(X)$. $\boldsymbol{w} = [p_T(\boldsymbol{x}_i)/p_S(\boldsymbol{x}_i)]^\top$.

To prove the, we only need to show the minimax linear estimator $A\boldsymbol{y}$ is achieved of the form $A_1 X^\top \operatorname{diag}(\boldsymbol{w})$, i.e., the row span of $A$ is in the row span of $X^\top \operatorname{diag}(\boldsymbol{w})$.

$$
\begin{aligned}
R_L(\mathcal{B}) &\equiv \min_A \max_{\boldsymbol{\beta}_T^* \in \mathcal{B}, a_T \in \mathcal{F}} \mathbb{E}_{\boldsymbol{x}_i \sim p_s, \boldsymbol{z}}[\|\Sigma_T^{1/2}(A\boldsymbol{y} - \boldsymbol{\beta}_T^*)\|^2] \\
&= \min_A \max_{\boldsymbol{\beta}_T^* \in \mathcal{B}, a_T \in \mathcal{F}} \mathbb{E} \|\Sigma_T^{1/2}((AX - I)\boldsymbol{\beta}_T^* + A\boldsymbol{a}_T + A\boldsymbol{z})\|^2
\end{aligned}
$$

$$\begin{aligned}
&= \min_A \max_{\boldsymbol{\beta}_T^* \in \mathcal{B}, a_T \in \mathcal{F}} \Big\{ \| \Sigma_T^{1/2}((\mathbb{E}[AX] - I)\boldsymbol{\beta}_T^* + \mathbb{E}[A\boldsymbol{a}_T]) \|_2^2 \\
&\quad + \mathbb{E}\, \| \Sigma_T^{1/2}(AX - \mathbb{E}[AX])\boldsymbol{\beta}_T^* \|^2 + \mathbb{E}\, \| \Sigma_T^{1/2}(A\boldsymbol{a}_T - \mathbb{E}[A\boldsymbol{a}_T]) \|^2 + \mathbb{E}\, \| \Sigma_T^{1/2} A\boldsymbol{z} \|^2 \Big\}
\end{aligned}$$

Write $A = A_1 X^\top \operatorname{diag}(\boldsymbol{w}) + A_2 W^\top$, where $X \in \mathbb{R}^{n \times d}$ and $W \in \mathbb{R}^{n \times (n-d)}$ forms the orthogonal complement for the column span of $\operatorname{diag}(\boldsymbol{w})X$. Therefore $X^\top \operatorname{diag}(\boldsymbol{w})W = 0$, and $W^\top W = I_{n-d}$. Also, notice $\mathbb{E}_{\boldsymbol{x}_i \sim p_S}[X^\top \operatorname{diag}(\boldsymbol{w})\boldsymbol{a}_T] = n\, \mathbb{E}_{\boldsymbol{x} \sim p_T}[\boldsymbol{x} a_T(\boldsymbol{x})] = 0$. Therefore plugging it in $R_L(\mathcal{B})$, we have:

$$\begin{aligned}
R_L(\mathcal{B}) &= \min_A \max_{\boldsymbol{\beta}_T^* \in \mathcal{B}, f^* \in \mathcal{F}} \Big\{ \| \Sigma_T^{1/2}((A_1 \mathbb{E}_{p_S}[X^\top \operatorname{diag}(\boldsymbol{w})X] - I)\boldsymbol{\beta}_T^* + A_2 \mathbb{E}[W^\top \boldsymbol{a}_T]) \|_2^2 \\
&\quad + \mathbb{E}\, \| \Sigma_T^{1/2} A_1 (X^\top \operatorname{diag}(\boldsymbol{w})X - \mathbb{E}[X^\top \operatorname{diag}(\boldsymbol{w})X])\boldsymbol{\beta}_T^* \|^2 \\
&\quad + \mathbb{E}\, \| \Sigma_T^{1/2} A_2 (W^\top \boldsymbol{a}_T - \mathbb{E}[W^\top \boldsymbol{a}_T]) \|^2 \\
&\quad + \sigma^2\, \mathbb{E}\, \| \Sigma_T^{1/2} A_1 X^\top \operatorname{diag}(\boldsymbol{w}) \|^2 + \sigma^2\, \mathbb{E}\, \| \Sigma_T^{1/2} A_2 \|^2 \Big\} \\
&= \min_{A_1, A_2} \max_{\boldsymbol{\beta}_T^* \in \mathcal{B}, f^* \in \mathcal{F}} \Big\{ \| \Sigma_T^{1/2}((A_1 n_S \Sigma_T - I)\boldsymbol{\beta}_T^* + A_2 \mathbb{E}[W^\top \boldsymbol{a}_T]) \|_2^2 \\
&\quad + \mathbb{E}\, \| \Sigma_T^{1/2} A_1 (X^\top \operatorname{diag}(\boldsymbol{w})X - \Sigma_T)\boldsymbol{\beta}_T^* \|^2 + \mathbb{E}\, \| \Sigma_T^{1/2} A_2 (W^\top \boldsymbol{a}_T - \mathbb{E}[W^\top \boldsymbol{a}_T]) \|^2 \\
&\quad + \sigma^2\, \mathbb{E}\, \| \Sigma_T^{1/2} A_1 X^\top \operatorname{diag}(\boldsymbol{w}) \|^2 + \sigma^2\, \mathbb{E}\, \| \Sigma_T^{1/2} A_2 \|^2 \Big\}
\end{aligned}$$

We could view $\mathbb{E}[W^\top \boldsymbol{a}_T]$ and $W^\top \boldsymbol{a}_T - \mathbb{E}[W^\top \boldsymbol{a}_T]$ separately. First notice at min-max point, if $\mathbb{E}[W^\top \boldsymbol{a}_T] = 0$, the minimizer $A_2$ should be 0 since it only appears in the third and last non-negative terms. If $\mathbb{E}[W^\top \boldsymbol{a}_T] \neq 0$, the cross term of the bias should be non-negative, or otherwise since both $f^*$ and $-f^*$ are in the set, $a_T, \boldsymbol{\beta}_T^*$ could be replaced by $-a_T, -\boldsymbol{\beta}_T^*$ and the loss increases. Clearly in this case $A_2$ should also be 0 at min-max point. □

### A.4 OMITTED PROOF FOR UTILIZING SOURCE AND TARGET DATA JOINTLY

**Sufficient statistic.**

*Proof of Claim 3.7.* Denote by $\bar{\boldsymbol{\beta}}_S := \hat{\Sigma}_S^{-1} X_S^\top \boldsymbol{y}_S / n_S \sim \mathcal{N}(\boldsymbol{\beta}^*, \frac{\sigma^2}{n_S}\hat{\Sigma}_S^{-1})$ and $\bar{\boldsymbol{\beta}}_T := \hat{\Sigma}_T^{-1} X_T^\top \boldsymbol{y}_T / n_T \sim \mathcal{N}(\boldsymbol{\beta}^*, \frac{\sigma^2}{n_T}\hat{\Sigma}_T^{-1})$. We use the Fisher–Neyman factorization theorem to derive the sufficient statistics. The likelihood of observing $\bar{\boldsymbol{\beta}}_S, \bar{\boldsymbol{\beta}}_T$ from parameter $\boldsymbol{\beta}^*$ is:

$$\begin{aligned}
p(\bar{\boldsymbol{\beta}}_S, \bar{\boldsymbol{\beta}}_T; \boldsymbol{\beta}^*) &= c e^{-\frac{n_S}{\sigma^2}(\bar{\boldsymbol{\beta}}_S - \boldsymbol{\beta}^*)\hat{\Sigma}_S(\bar{\boldsymbol{\beta}}_S - \boldsymbol{\beta}^*) - \frac{n_T}{\sigma^2}(\bar{\boldsymbol{\beta}} - \boldsymbol{\beta}^*)\bar{\Sigma}_T(\bar{\boldsymbol{\beta}}_T - \boldsymbol{\beta}^*)} \\
&= c g(\boldsymbol{\beta}^*, T(\boldsymbol{\beta}^*)) h(\bar{\boldsymbol{\beta}}_S, \bar{\boldsymbol{\beta}}_T),
\end{aligned}$$

where $g(\boldsymbol{\beta}^*, T(\boldsymbol{\beta}^*)) = e^{-(\boldsymbol{\beta}^* - \hat{\boldsymbol{\beta}}_{\text{SS}})^\top (\frac{n_S}{\sigma^2}\hat{\Sigma}_S + \frac{n_T}{\sigma^2}\hat{\Sigma}_T)^{-1}(\boldsymbol{\beta}^* - \hat{\boldsymbol{\beta}}_{\text{SS}})}$, and $c$ is some constant. Therefore it's easy to see that $T(\boldsymbol{\beta}^*) = \hat{\boldsymbol{\beta}}_{\text{SS}}$ is the sufficient statistic for $\boldsymbol{\beta}^*$. □

*Proof of Claim 3.8.* With similar procedure as before, and notice $\boldsymbol{z}_S$ and $\boldsymbol{z}_T$ are independent, we could first conclude that the optimal estimator is of the form $\hat{\boldsymbol{\beta}} = A\hat{\Sigma}_S^{-1} X_S^\top \boldsymbol{y}_S / n_S + B\hat{\Sigma}_T^{-1} X_T^\top \boldsymbol{y}_T / n_T \sim \mathcal{N}((A+B)\boldsymbol{\beta}^*, \frac{\sigma^2}{n_S} A\hat{\Sigma}_S^{-1} A^\top + \frac{\sigma^2}{n_T} B\hat{\Sigma}_T^{-1} B^\top)$.

$$R_L(\mathcal{B}) = \min_{A, B} \max_{\boldsymbol{\beta}^* \in \mathcal{B}} \mathbb{E}_{\boldsymbol{z}}\, \| \Sigma_T^{1/2}(\hat{\boldsymbol{\beta}} - \boldsymbol{\beta}^*) \|^2$$

$$= \min_{A,B} \max_{\boldsymbol{\beta}^* \in \mathcal{B}} \left\{ \|\Sigma^{1/2}(A + B - I)\boldsymbol{\beta}^*\|^2 \right.$$

$$\left. + \sigma^2 \text{Tr}((\frac{1}{n_S} A \hat{\Sigma}_S^{-1} A^\top + \frac{1}{n_T} B \hat{\Sigma}_T^{-1} B^\top)\Sigma_T) \right\}$$

$$= \min_{A,B} \left\{ \|\Sigma^{1/2}(A + B - I)\|_{op}^2 r^2 + \sigma^2 \text{Tr}((\frac{1}{n_S} A \hat{\Sigma}_S^{-1} A^\top + \frac{1}{n_T} B \hat{\Sigma}_T^{-1} B^\top)\Sigma_T) \right\}$$

Take gradient w.r.t $A$ and $B$ respectively we have:

$$\nabla_A(\|\Sigma^{1/2}(A + B - I)\|_{op}^2 r^2) + \frac{\sigma^2}{n_S} \Sigma_T A \hat{\Sigma}_S^{-1} = 0$$

$$= \nabla_B(\|\Sigma^{1/2}(A + B - I)\|_{op}^2 r^2) + \frac{\sigma^2}{n_T} \Sigma_T B \hat{\Sigma}_T^{-1} = 0$$

Notice the first terms are equivalent. Therefore $\frac{1}{n_S} A \hat{\Sigma}_S^{-1} = \frac{1}{n_T} B \hat{\Sigma}_T^{-1}$ thus the optimal $\hat{\boldsymbol{\beta}}$ is of the form $C(X_S^\top \boldsymbol{y}_S + X_T^\top \boldsymbol{y}_T)$ for some matrix $C$, thus finishing the proof. $\qquad\square$

## B  OMITTED PROOF WITH MODEL SHIFT

**Definition B.1** (Orthosymmetry). *A set $\Theta$ is said to be solid and orthosymmetric if $\boldsymbol{\theta} \in \Theta$ and $|\zeta_i| \leq |\theta_i|$ for all $i$ implies that $\boldsymbol{\zeta} \in \Theta$. If a solid, orthosymmetric $\Theta$ contains a point $\boldsymbol{\tau}$, then it contains the entire hyperrectangle that $\boldsymbol{\tau}$ defines: $\Theta(\boldsymbol{\tau}) \equiv \{\boldsymbol{\theta} | |\theta_i| \leq \tau_i, \forall i\} \subset \Theta$.*

*Proof of Claim 4.1.* First notice for any estimator $\hat{\boldsymbol{\beta}}$, it all satisfies

$$L_{\mathcal{B},\Delta}(\hat{\boldsymbol{\beta}}) \leq r_{\mathcal{B},\Delta}(\hat{\boldsymbol{\beta}}) \leq 2L_{\mathcal{B},\Delta}(\hat{\boldsymbol{\beta}}). \tag{9}$$

The first inequality is straightforward with the same reasoning of AM-GM as the derivation of (5). As for the second inequality, we take a closer look at (5). Notice that when $\max_{\boldsymbol{\beta}_T^* \in \mathcal{B}, \boldsymbol{\delta} \in \Delta}$ is achieved, the cross term has to be non-negative, or otherwise one could flip the sign of $\boldsymbol{\beta}_T^*$ to make the value larger. Therefore at maximum $\|\Sigma_T^{1/2}((A_1 + A_2 - I)\boldsymbol{\beta}_T^*\|^2 + \|\Sigma_T^{1/2} A_1 \boldsymbol{\delta}\|^2 \leq \|\Sigma_T^{1/2}((A_1 + A_2 - I)\boldsymbol{\beta}_T^* + \Sigma_T^{1/2} A_1 \boldsymbol{\delta}\|^2$, and notice the remaining parts are all non-negative. Therefore $r_{\mathcal{B},\Delta}(\hat{\boldsymbol{\beta}}) \leq 2L_{\mathcal{B},\Delta}(\hat{\boldsymbol{\beta}})$.

Now let $\hat{\boldsymbol{\beta}}^* = \arg\min_{\hat{\boldsymbol{\beta}} = A_1 \bar{\boldsymbol{y}}_S + A_2 \bar{\boldsymbol{y}}_S} L_{\mathcal{B},\Delta}(\hat{\boldsymbol{\beta}})$. We have:

$$R_L(\mathcal{B}, \Delta) = L_{\mathcal{B},\Delta}(\hat{\boldsymbol{\beta}}^*) \overset{(a)}{\leq} L_{\mathcal{B},\Delta}(\hat{\boldsymbol{\beta}}_{\text{MM}})$$

$$\overset{(9)}{\leq} r_{\mathcal{B},\Delta}(\hat{\boldsymbol{\beta}}_{\text{MM}}) \overset{(b)}{\leq} r_{\mathcal{B},\Delta}(\hat{\boldsymbol{\beta}}^*) \overset{(9)}{\leq} 2L_{\mathcal{B},\Delta}(\hat{\boldsymbol{\beta}}^*) = 2R_L(\mathcal{B}, \Delta).$$

The inequality (a) is by definition of $\hat{\boldsymbol{\beta}}^*$ while (b) is from the definition of $\hat{\boldsymbol{\beta}}_{\text{MM}}$. $\qquad\square$

### B.1  LOWER BOUND WITH MODEL SHIFT

In order to derive the lower bound, we abstract the problem to the following more general one:

**Problem 1.** *For arbitrary diagonal matrix $D \in \mathbb{R}^{d \times d}$, two $\ell_2$-compact, solid, orthosymmetric, and quadratically convex sets $\Theta, \Delta \subset \mathbb{R}^d$, let*

$$\mathcal{P}_{\Theta, \Delta, D} = \left\{ \mathcal{N}\left( \begin{bmatrix} D\boldsymbol{\theta} + \boldsymbol{\delta} \\ \boldsymbol{\theta} \end{bmatrix}, \begin{bmatrix} I & 0 \\ 0 & I \end{bmatrix} \right) \Big| \boldsymbol{\theta} \in \Theta, \boldsymbol{\delta} \in \Delta \right\}$$

*Let $R_L(\Theta, \Delta, D)$ and $R_N(\Theta, \Delta, D)$ be the minimax linear risk and minimax risk respectively for estimating $\boldsymbol{\theta}$ within the distribution class $\mathcal{P}_{\Theta, \Delta, D}$:*

$$R_L(\Theta, \Delta, D) = \min_{\hat{\boldsymbol{\theta}}: \mathbb{R}^d \to \Theta \ linear} \max_{P \in \mathcal{P}_{\Theta, \Delta, D}} r_P(\hat{\boldsymbol{\theta}}),$$

$$R_N(\Theta, \Delta, D) = \min_{\hat{\boldsymbol{\theta}}: \mathbb{R}^d \to \Theta} \max_{P \in \mathcal{P}_{\Theta, \Delta, D}} r_P(\hat{\boldsymbol{\theta}}).$$

*Here $r_P(\hat{\boldsymbol{\theta}}) := \mathbb{E}_{\boldsymbol{x} \sim P} \|\hat{\boldsymbol{\theta}}(\boldsymbol{x}) - \boldsymbol{\theta}(P)\|_2^2$. We want to derive a uniform lower bound for $R_N$ with $R_L$, i.e., $R_N \geq \mu^* R_L$, where $\mu^*$ is universal and doesn't depend on the choices of $D$, $\Theta$ or $\Delta$.*

Before proving the lower bound, we establish its connection to our considered problem:

**Remark B.1.** *Suppose $\Sigma_S = U\text{diag}(\boldsymbol{s})U^\top$ and $\Sigma_T = U\text{diag}(\boldsymbol{t})U^\top$ share the same eigenspace. Recall our samples $\boldsymbol{a} \sim \mathcal{N}(\Sigma_S^{1/2}(\boldsymbol{\beta}_T^* + \boldsymbol{\delta}), \sigma^2 I), \boldsymbol{b} \sim \mathcal{N}(\Sigma_T^{1/2}\boldsymbol{\beta}_T^*, \sigma^2 I)$. Our goal to uniformly lower bound $R_N(r, \gamma)$ by $R_L(r, \gamma)$ is essentially Problem 1, where*

$$R_L(r, \gamma) := \min_{\hat{\boldsymbol{\beta}} \ linear} \max_{\|\boldsymbol{\beta}_T^*\| \leq r, \|\boldsymbol{\delta}\| \leq \gamma} \mathbb{E} \|\Sigma_T^{1/2}(\hat{\boldsymbol{\beta}}(\boldsymbol{a}, \boldsymbol{b}) - \boldsymbol{\beta}^*)\|^2,$$

$$R_N(r, \gamma) := \min_{\hat{\boldsymbol{\beta}}} \max_{\|\boldsymbol{\beta}_T^*\| \leq r, \|\boldsymbol{\delta}\| \leq \gamma} \mathbb{E} \|\Sigma_T^{1/2}(\hat{\boldsymbol{\beta}}(\boldsymbol{a}, \boldsymbol{b}) - \boldsymbol{\beta}^*)\|^2.$$

*Proof of Remark B.1.* Our target considers samples drawn from distributions $\boldsymbol{x} \sim \mathcal{N}(\Sigma_S^{1/2}(\boldsymbol{\beta}_T^* + \boldsymbol{\delta}), \sigma^2 I), \boldsymbol{y} \sim \mathcal{N}(\Sigma_T^{1/2}\boldsymbol{\beta}_T^*, \sigma^2 I)$.

$$\begin{bmatrix} \boldsymbol{a} \\ \boldsymbol{b} \end{bmatrix} \sim \mathcal{N}\left( \begin{bmatrix} U\text{diag}(\boldsymbol{s}^{1/2})U^\top(\boldsymbol{\beta}_T^* + \boldsymbol{\delta}) \\ U\text{diag}(\boldsymbol{t}^{1/2})U^\top\boldsymbol{\beta}_T^* \end{bmatrix}, \begin{bmatrix} \sigma^2 I & 0 \\ 0 & \sigma^2 I \end{bmatrix} \right), \boldsymbol{\theta} \in \Theta, \boldsymbol{\delta} \in \Delta$$

$$\iff \begin{bmatrix} U^\top\boldsymbol{a}/\sigma \\ U^\top\boldsymbol{b}/\sigma \end{bmatrix} \sim \mathcal{N}\left( \begin{bmatrix} \text{diag}(\boldsymbol{s}^{1/2})U^\top(\boldsymbol{\beta}_T^* + \boldsymbol{\delta}) \\ \text{diag}(\boldsymbol{t}^{1/2})U^\top\boldsymbol{\beta}_T^* \end{bmatrix}, \begin{bmatrix} I & 0 \\ 0 & I \end{bmatrix} \right), \|\boldsymbol{\beta}_T^*\| \leq r, \|\boldsymbol{\delta}\| \in \gamma$$

Let $\bar{\boldsymbol{a}} = U^\top\boldsymbol{a}/\sigma, \bar{\boldsymbol{b}} = U^\top\boldsymbol{b}/\sigma, \Theta = \{\boldsymbol{\theta} | \|\text{diag}(\boldsymbol{t}^{-1/2})\boldsymbol{\theta}\| \leq r\}, \Delta = \{\|\text{diag}(\boldsymbol{s}^{-1/2})\boldsymbol{\delta}\| \leq \gamma\}$. $\bar{\boldsymbol{\theta}} = U^\top\Sigma_T^{1/2}\boldsymbol{\beta}_T^*, \bar{\boldsymbol{\delta}} = U^\top\Sigma_S^{1/2}\boldsymbol{\delta}$, and $D = \text{diag}(\boldsymbol{s}^{1/2}\boldsymbol{t}^{-1/2})$. We get:

$$\begin{bmatrix} U^\top\boldsymbol{a}/\sigma \\ U^\top\boldsymbol{b}/\sigma \end{bmatrix} \sim \mathcal{N}\left( \begin{bmatrix} \text{diag}(\boldsymbol{s}^{1/2})U^\top(\boldsymbol{\beta}_T^* + \boldsymbol{\delta}) \\ \text{diag}(\boldsymbol{t}^{1/2})U^\top\boldsymbol{\beta}_T^* \end{bmatrix}, \begin{bmatrix} I & 0 \\ 0 & I \end{bmatrix} \right), \|\boldsymbol{\beta}_T\| \leq r, \|\boldsymbol{\delta}\| \in \gamma$$

$$\iff \begin{bmatrix} \bar{\boldsymbol{a}} \\ \bar{\boldsymbol{b}} \end{bmatrix} \sim P_{\boldsymbol{\theta}, \boldsymbol{\delta}, D} := \mathcal{N}\left( \begin{bmatrix} D\bar{\boldsymbol{\theta}} + \bar{\boldsymbol{\delta}} \\ \bar{\boldsymbol{\theta}} \end{bmatrix}, \begin{bmatrix} I & 0 \\ 0 & I \end{bmatrix} \right), \bar{\boldsymbol{\theta}} \in \Theta, \bar{\boldsymbol{\delta}} \in \Delta.$$

Let $\mathcal{P}_{\Theta, \Delta, D} := \{P_{\bar{\boldsymbol{\theta}}, \bar{\boldsymbol{\delta}}, D} | \bar{\boldsymbol{\theta}} \in \Theta, \bar{\boldsymbol{\delta}} \in \Delta\}$. Since $U$ is an invertible matrices, observing $U^\top\boldsymbol{a}/\sigma, U^\top\boldsymbol{b}/\sigma$ instead of $\boldsymbol{a}, \boldsymbol{b}$ has no affect on the performance of the best estimator. Also $\Theta, \Delta$ are axis-aligned ellipsoid and thus satisfy orthosymmetry. Therefore our problem is essentially reduced to Problem 1. $\square$

**Lemma B.2.** *Let $\Theta(\boldsymbol{\tau}) = \{\boldsymbol{\theta} | \theta_i \leq \tau_i, \forall i, \boldsymbol{\theta} \in \Theta\}$ and similarly for $\Delta(\boldsymbol{\zeta}) = \{\boldsymbol{\delta} | \delta_i \leq \zeta_i, \boldsymbol{\delta} \in \Delta\}$, $D$ is some diagonal matrix.*

$$R_L(\Theta, \Delta, D) = \sup_{\boldsymbol{\tau} \in \Theta, \boldsymbol{\zeta} \in \Delta} R_L(\Theta(\boldsymbol{\tau}), \Delta(\boldsymbol{\zeta}), D), \text{ and}$$

$$R_N(\Theta, \Delta, D) \geq \sup_{\boldsymbol{\tau} \in \Theta, \boldsymbol{\zeta} \in \Delta} R_N(\Theta(\boldsymbol{\tau}), \Delta(\boldsymbol{\zeta}), D).$$

Write samples drawn from some $P_{\boldsymbol{\theta}, \boldsymbol{\delta}, D} \in \mathcal{P}_{\Theta, \Delta, D}$ as $(\boldsymbol{x}, \boldsymbol{y}) : \boldsymbol{x} \sim \mathcal{N}(D\boldsymbol{\theta} + \boldsymbol{\delta}, I), \boldsymbol{y} \sim \mathcal{N}(\boldsymbol{\theta}, I)$.

**Lemma B.3.** *The minimax linear estimator $\hat{\boldsymbol{\theta}} : (\boldsymbol{x}, \boldsymbol{y}) \to A\boldsymbol{x} + B\boldsymbol{y}$ has the form $\hat{\boldsymbol{\theta}}_{\boldsymbol{a}, \boldsymbol{b}}(\boldsymbol{x}, \boldsymbol{y}) = \sum_i a_i x_i + \sum_i b_i y_i$ for some $\boldsymbol{a}, \boldsymbol{b} \in \mathbb{R}^d$. Namely,*

$$R_L(\Theta, \Delta, D) = \inf_{\hat{\boldsymbol{\theta}}_{\boldsymbol{a}, \boldsymbol{b}}} \max_{P \in \mathcal{P}_{\Theta, \Delta, D}} r_P(\hat{\boldsymbol{\theta}}_{\boldsymbol{a}, \boldsymbol{b}}).$$

*Proof.* According to the proof of Proposition B.4.a, by discarding off-diagonal terms, the maximum risk of any linear estimator $\hat{\boldsymbol{\theta}}_{A,B}$ over any hyperrectangles $\Theta(\boldsymbol{\tau}), \Delta(\boldsymbol{\zeta})$ is reduced.

$$\max_{\boldsymbol{\theta} \in \Theta(\boldsymbol{\tau}), \boldsymbol{\delta} \in \Delta(\boldsymbol{\zeta})} r_{P_{\boldsymbol{\theta}, \boldsymbol{\delta}, D}}(\hat{\boldsymbol{\theta}}_{A,B}) \geq \max_{\boldsymbol{\theta} \in \Theta(\boldsymbol{\tau}), \boldsymbol{\delta} \in \Delta(\boldsymbol{\zeta})} r_{P_{\boldsymbol{\theta}, \boldsymbol{\delta}, D}}(\hat{\boldsymbol{\theta}}_{\mathrm{diag}(A), \mathrm{diag}(B)}).$$

Further we have:

$$\min_{A,B} \max_{\boldsymbol{\theta} \in \Theta, \boldsymbol{\delta} \in \Delta} r_{P_{\boldsymbol{\theta}, \boldsymbol{\delta}, D}}(\hat{\boldsymbol{\theta}}_{A,B}) \geq \min_{A,B} \max_{\boldsymbol{\tau} \in \Theta, \boldsymbol{\zeta} \in \Delta} \max_{\boldsymbol{\theta} \in \Theta(\boldsymbol{\tau}), \boldsymbol{\delta} \in \Delta(\boldsymbol{\zeta})} r_{P_{\boldsymbol{\theta}, \boldsymbol{\delta}, D}}(\hat{\boldsymbol{\theta}}_{\mathrm{diag}(A), \mathrm{diag}(B)})$$

$$= \min_{\boldsymbol{a}, \boldsymbol{b}} \max_{\boldsymbol{\theta} \in \Theta, \boldsymbol{\zeta} \in \Delta} r_{P_{\boldsymbol{\theta}, \boldsymbol{\delta}, D}}(\hat{\boldsymbol{\theta}}_{\boldsymbol{a}, \boldsymbol{b}})$$

$$\geq \min_{C} \max_{\boldsymbol{\theta} \in \Theta, \boldsymbol{\delta} \in \Delta} r_{P_{\boldsymbol{\theta}, \boldsymbol{\delta}, D}}(\hat{\boldsymbol{\theta}}_{A,B}).$$

Therefore all four terms have to be equal, thus finishing the proof. □

Notice $\Theta(\boldsymbol{\tau})$ and $\Delta(\boldsymbol{\zeta})$ are hyperrectangles in $\mathbb{R}^d$. Therefore we could decompose the problem to some 2-d problems:

**Proposition B.4.** *Under the same setting as Problem 1,*

$$a). \ R_L(\Theta(\boldsymbol{\tau}), \Delta(\boldsymbol{\zeta}), D) = \sum_i R_L(\tau_i, \zeta_i, D_{ii}).$$

*If $\hat{\boldsymbol{\theta}}_{A,B}(\boldsymbol{x}, \boldsymbol{y}) = A\boldsymbol{x} + B\boldsymbol{y}$ is minimax linear estimator over $P_{\Theta(\boldsymbol{\tau}), \Delta(\boldsymbol{\zeta}), D}$, then necessarily $A, B$ must be diagonal.*

$$b). \ R_N(\Theta(\boldsymbol{\tau}), \Delta(\boldsymbol{\zeta}), D) = \sum_i R_N(\tau_i, \zeta_i, D_{ii}).$$

*Proof of Proposition B.4.a .* First review our notation:

$$\begin{aligned} r_{P_{\boldsymbol{\theta}, \boldsymbol{\delta}, D}}(\hat{\boldsymbol{\theta}}_{A,B}) &= \mathbb{E}_{(\boldsymbol{x}, \boldsymbol{y}) \sim P_{\boldsymbol{\theta}, \boldsymbol{\delta}, D}} \|\hat{\boldsymbol{\theta}}_{A,B}(\boldsymbol{x}, \boldsymbol{y}) - \boldsymbol{\theta}\|^2 \\ &= \mathbb{E}_{\boldsymbol{x} \sim \mathcal{N}(D\boldsymbol{\theta} + \boldsymbol{\delta}, I), \boldsymbol{y} \sim \mathcal{N}(\boldsymbol{\theta}, I)} \|A\boldsymbol{x} + B\boldsymbol{y} - \boldsymbol{\theta}\|^2 \\ &= \|A(D\boldsymbol{\theta} + \boldsymbol{\delta}) + B\boldsymbol{\theta} - \boldsymbol{\theta}\|^2 + \mathrm{Tr}(AA^\top) + \mathrm{Tr}(BB^\top) \\ &= \|(AD + B - I)\boldsymbol{\theta} + A\boldsymbol{\delta}\|^2 + \mathrm{Tr}(AA^\top) + \mathrm{Tr}(BB^\top). \end{aligned}$$

Our objective is

$$R_L(\Theta(\boldsymbol{\tau}), \Delta(\boldsymbol{\zeta}), D) := \min_{A,B} \max_{\boldsymbol{\theta} \in \Theta(\boldsymbol{\tau}), \boldsymbol{\delta} \in \Delta(\boldsymbol{\zeta})} r_{P_{\boldsymbol{\theta}, \boldsymbol{\delta}, D}}(\hat{\boldsymbol{\theta}}_{A,B})$$

We will show that restricting $A, B$ to be diagonal will not include the RHS value.

For any $\bar{\boldsymbol{\tau}} \in \Theta(\boldsymbol{\tau}), \bar{\boldsymbol{\zeta}} \in \Delta(\boldsymbol{\zeta})$, let set $V(\bar{\boldsymbol{\tau}}, \bar{\boldsymbol{\zeta}}) = \{(\boldsymbol{\theta}, \boldsymbol{\delta}) | (\theta_i, \delta_i) \in \{(\bar{\tau}_i, \bar{\zeta}_i), (-\bar{\tau}_i, -\bar{\zeta}_i)\}\}$ be the subset of vertices of $\Theta(\bar{\boldsymbol{\tau}}) \times \Delta(\bar{\boldsymbol{\zeta}})$. Let $\pi(\bar{\boldsymbol{\tau}}, \bar{\boldsymbol{\zeta}})$ be uniform distribution on this finite set. Due to the symmetry of this distribution, we have

$$\mathbb{E}_{\pi(\bar{\boldsymbol{\tau}}, \bar{\boldsymbol{\zeta}})} \theta_i = 0, i \in [d],$$

$$\mathbb{E}_{\pi(\bar{\boldsymbol{\tau}}, \bar{\boldsymbol{\zeta}})} \delta_i = 0, i \in [d],$$

$$\mathbb{E}_{\pi(\bar{\boldsymbol{\tau}}, \bar{\boldsymbol{\zeta}})} \theta_i \theta_j = \mathbf{1}_{i=j} \bar{\tau}_i^2, i \in [d],$$

$$\mathbb{E}_{\pi(\bar{\boldsymbol{\tau}}, \bar{\boldsymbol{\zeta}})} \delta_i \delta_j = \mathbf{1}_{i=j} \bar{\zeta}_i^2, i \in [d],$$

$$\mathbb{E}_{\pi(\bar{\boldsymbol{\tau}}, \bar{\boldsymbol{\zeta}})} \theta_i \delta_j = \mathbf{1}_{i=j} \bar{\tau}_i \bar{\zeta}_i, i \in [d].$$

We utilize the distribution to find the explicit value of the maximum (in fact the maximum will only be obtained inside the vertices set $V(\bar{\boldsymbol{\tau}}, \bar{\boldsymbol{\zeta}})$ ):

$$\max_{(\boldsymbol{\theta}, \boldsymbol{\delta}) \in V(\bar{\boldsymbol{\tau}}, \bar{\boldsymbol{\zeta}})} r_{P_{\boldsymbol{\theta}, \boldsymbol{\delta}, D}}(\hat{\boldsymbol{\theta}}_{A,B}) \geq \mathbb{E}_{\pi(\bar{\boldsymbol{\tau}}, \bar{\boldsymbol{\zeta}})} r_{P_{\boldsymbol{\theta}, \boldsymbol{\delta}, D}}(\hat{\boldsymbol{\theta}}_{A,B})$$

$$= \mathbb{E}_{\pi(\bar{\boldsymbol{\tau}}, \bar{\boldsymbol{\zeta}})} \|(AD + B - I)\boldsymbol{\theta} + A\boldsymbol{\delta}\|^2 + \text{Tr}(AA^\top) + \text{Tr}(BB^\top)$$

$$= \text{Tr}((AD + B - I) \mathbb{E}[\boldsymbol{\theta}\boldsymbol{\theta}^\top](AD + B - I)^\top) + \text{Tr}(A \mathbb{E}[\boldsymbol{\delta}\boldsymbol{\delta}^\top]A^\top) +$$
$$\quad 2\text{Tr}((AD + B - I) \mathbb{E}[\boldsymbol{\theta}\boldsymbol{\delta}^\top]A^\top) + \text{Tr}(AA^\top) + \text{Tr}(BB^\top)$$

$$= \text{Tr}((AD + B - I)^\top (AD + B - I)\text{diag}(\bar{\boldsymbol{\tau}}^2)) + \text{Tr}(A^\top A \text{diag}(\bar{\boldsymbol{\zeta}}^2))$$
$$\quad + \text{Tr}((AD + B - I)^\top A \text{diag}(\bar{\boldsymbol{\tau}}\bar{\boldsymbol{\zeta}})) + \text{Tr}(AA^\top) + \text{Tr}(BB^\top)$$

$$= \sum_i \|(AD + B - I)_{:,i} \bar{\tau}_i + A_{:,i} \bar{\zeta}_i\|^2 + \text{Tr}(AA^\top) + \text{Tr}(BB^\top)$$

$$\geq \sum_i ((A_{ii}D_{ii} + B_{ii} - 1)\bar{\tau}_i + A_{ii}\bar{\zeta}_i)^2 + A_{ii}^2 + B_{ii}^2$$

$$= \|(\text{diag}(A)D + \text{diag}(B) - I)\boldsymbol{\theta} + \text{diag}(A)\boldsymbol{\delta}\|^2 + \text{Tr}(\text{diag}(A)^2) + \text{Tr}(\text{diag}(B)^2),$$
$$\qquad\qquad\qquad\qquad (\forall (\boldsymbol{\theta}, \boldsymbol{\delta}) \in V(\bar{\boldsymbol{\tau}}, \bar{\boldsymbol{\zeta}}))$$

$$= \max_{V(\bar{\boldsymbol{\tau}}, \bar{\boldsymbol{\zeta}})} \|(\text{diag}(A)D + \text{diag}(B) - I)\boldsymbol{\theta} + \text{diag}(A)\boldsymbol{\delta}\|^2 + \text{Tr}(\text{diag}(A)^2) + \text{Tr}(\text{diag}(B)^2)$$

Therefore we have:

$$R_L(\Theta(\boldsymbol{\tau}), \Delta(\boldsymbol{\zeta}), D) := \min_{A,B} \max_{\boldsymbol{\theta} \in \Theta(\boldsymbol{\tau}), \boldsymbol{\delta} \in \Delta(\boldsymbol{\zeta})} r_{P_{\boldsymbol{\theta}, \boldsymbol{\delta}, D}}(\hat{\boldsymbol{\theta}}_{A,B})$$

$$= \min_{A,B} \max_{\bar{\boldsymbol{\tau}} \in \Theta(\boldsymbol{\tau}), \bar{\boldsymbol{\zeta}} \in \Delta(\boldsymbol{\zeta})} \max_{\boldsymbol{\theta} \in V(\bar{\boldsymbol{\tau}}, \bar{\boldsymbol{\zeta}})} r_{P_{\boldsymbol{\theta}, \boldsymbol{\delta}, D}}(\hat{\boldsymbol{\theta}}_{A,B})$$

$$\geq \min_{A,B} \max_{\bar{\boldsymbol{\tau}} \in \Theta(\boldsymbol{\tau}), \bar{\boldsymbol{\zeta}} \in \Delta(\boldsymbol{\zeta})} \max_{(\boldsymbol{\theta}, \boldsymbol{\delta}) \in V(\bar{\boldsymbol{\tau}}, \bar{\boldsymbol{\zeta}})} r_{P_{\boldsymbol{\theta}, \boldsymbol{\delta}, D}}(\hat{\boldsymbol{\theta}}_{\text{diag}(A), \text{diag}(B)})$$

$$= \min_{\boldsymbol{a} \in \mathbb{R}^d, \boldsymbol{b} \in \mathbb{R}^d} \max_{\boldsymbol{\theta} \in \Theta(\boldsymbol{\tau}), \boldsymbol{\delta} \in \Delta(\boldsymbol{\zeta})} r_{P_{\boldsymbol{\theta}, \boldsymbol{\delta}, D}}(\hat{\boldsymbol{\theta}}_{\boldsymbol{a}, \boldsymbol{b}}).$$

Next, since the optimal solution on the minimizer is always obtained by diagonal $A, B$, it becomes straightforward that each axis could be viewed in separation, thus finishing the proof for part a.

The nonlinear part is a straightforward extension of Proposition 4.16 from Johnstone (2011).

$\square$

**Theorem B.5** (Restated Le Cam Two Point Theorem Wainwright (2019)). *Let $\mathcal{P}$ be a family of distribution, and $\theta : \mathcal{P} \to \Theta$ is some associated parameter. Let $\rho : \Theta \times \Theta \to \mathbb{R}^+$ be some metric defined on $\Theta$ and $\Phi : \mathbb{R}_+ \to \mathbb{R}_+$ is a monotone non-decreasing function with $\Phi(0) = 0$. For any $\alpha \in (0, 1)$,*

$$\inf_{\hat{\theta}} \sup_{P \in \mathcal{P}} [\Phi(\rho(\hat{\theta}, \theta(P)))] \geq \max_{P_1, P_2 \in \mathcal{P}} \frac{1}{2} \Phi(\frac{1}{2} \rho(\theta(P_1), \theta(P_2)))(1 - \alpha),$$

$$s.t. \ \|P_1^n - P_2^n\|_{TV} \leq \alpha.$$

**Lemma B.6.** *Consider a class of distribution $\mathcal{P}_{\tau, \zeta, s} = \{P_{\theta, \delta, s} | P_{\theta, \delta, s} := \mathcal{N}([s\theta + \delta, \theta]^\top, I_2), |\theta| \leq \tau, |\delta| \leq \zeta\}$. Define*

$$R_L(\tau, \zeta, s) = \min_{\hat{\theta} \ linear} \max_{|\theta| \leq \tau, |\delta| \leq \zeta} \mathbb{E}_{\boldsymbol{x} \sim P_{\theta, \delta, s}} (\hat{\theta}(\boldsymbol{x}) - \theta)^2,$$

$$and \ R_N(\tau, \zeta, s) = \min_{\hat{\theta}} \max_{|\theta| \leq \tau, |\delta| \leq \zeta} \mathbb{E}_{\boldsymbol{x} \sim P_{\theta, \delta, s}} (\hat{\theta}(\boldsymbol{x}) - \theta)^2$$

*We have*

$$R_L(\tau, \zeta, s) \leq 27/2 R_N(\tau, \zeta, s), \forall \zeta, s > 0, \tau > 0.$$

*Proof of Lemma B.6.* We first calculate an upper bound of $R_L$ and connect it to a lower bound of $R_N$.

$$R_L(\tau, \zeta, s) = \min_{a, b} \max_{|\theta| \leq \tau, |\delta| \leq \zeta} [(as + b - 1)\theta + a\delta]^2 + a^2 + b^2$$

$$= \min_{a, b} (|as + b - 1|\tau + |a|\zeta)^2 + a^2 + b^2$$

$$\leq \min_{a, b} 2(as + b - 1)^2 \tau^2 + 2a^2 \zeta^2 + a^2 + b^2.$$

By some detailed calculations, we get the RHS is equal to:

$$\frac{2\tau^2(2\zeta^2 + 1)}{2\tau^2(s^2 + 2\zeta^2 + 1) + 2\zeta^2 + 1}$$

$$\leq \min\{1, 2\tau^2, \frac{1 + 4\zeta^2}{s^2 + 1}\}.$$

For simplify this form, we could see that

Next, we use Le cam two point theorem to lower bound $R_N(\tau, \zeta, s)$ where the metric $\rho$ is Euclidean distance and $\Phi$ is squared function. Therefore

$$R_N(\tau, \zeta, s) \geq \max_{|\theta_i| \leq \tau, |\delta_i| \leq \zeta, i \in \{1, 2\}} \frac{1}{2} (\frac{1}{2} (\theta_1 - \theta_2))^2 (1 - \alpha)$$

$$s.t. \ \|\mathcal{N}([s\theta_1 + \delta_1, \theta_1]^\top, I_2), \mathcal{N}([s\theta_2 + \delta_2, \theta_2]^\top, I_2)\|_{TV} \leq \alpha.$$

Since the total variation distance is related to Kullback-Leibler divergence by Pinsker's inequality: $\|\cdot,\cdot\|_{TV} \leq \sqrt{\frac{1}{2}D_{KL}(\cdot\|\cdot)}$, it's sufficient to replace the constraint as:

$$D_{KL}\left(\mathcal{N}([s\theta_1+\delta_1,\theta_1]^\top, I_2) \,\|\, \mathcal{N}([s\theta_2+\delta_2,\theta_2]^\top, I_2)\right) \leq 2\alpha^2.$$

$$\max_{|\theta_i|\leq\tau,|\delta_i|\leq\zeta,i\in\{1,2\}} \frac{1}{8}(\theta_1-\theta_2)^2(1-\alpha)$$
$$\text{s.t. } (s\theta_1+\delta_1-(s\theta_2+\delta_2))^2+(\theta_1-\theta_2)^2 \leq 2\alpha^2$$
$$\Leftrightarrow \max_{|c|\leq 2\tau,|d|\leq 2\zeta} \frac{c^2}{8}(1-\alpha)$$
$$\text{s.t. } (sc+d)^2+c^2 \leq 2\alpha^2.$$

Recall $R_L \leq \min\{1, 2\tau^2, \frac{1+4\zeta}{s^2+1}\}$.

We first note that $c^2 \leq 4\tau^2$ and setting $\alpha=0$ we have $R_N \geq \tau^2/2 \geq 1/4R_L$. For In the following we look at other cases when the bound for $c^2$ is smaller.

When $2\zeta \geq sc$, will set $d=-sc$ and $c^2=2\alpha^2$. Let $\alpha=2/3$ for large $\tau$ we get : $c^2(1-\alpha)/8 = 2/27 \geq 2/27R_L$.

When $2\zeta \leq sc$ we set $d=-2\zeta$ and require $(sc-2\zeta)^2+c^2 \leq 2\alpha^2$. We have $(sc-2\zeta)^2+c^2 = s^2c^2+4\zeta^2-4\zeta sc+c^2 \leq s^2c^2+4\zeta^2-8\zeta^2+c^2 = (s^2+1)c^2-4\zeta^2$. Therefore as we set $c^2 = \frac{2\alpha^2+4\zeta^2}{s^2+1}$, the original inequality is satisfied. Again by setting $\alpha=2/3$ we have $c^2 \geq 8/9\frac{1+4\zeta^2}{s^2+1} \geq 8/9R_L$. Therefore in this case $R_N \geq \frac{2}{27}R_L$.

$\square$

## C  DISCUSSIONS ON RANDOM DESIGN UNDER COVARIATE SHIFT.

In the main text, we present the results where we consider $X_S$ as fixed and $\Sigma_T$ to be known. In this section, we view both source and target input data as random, and generalize the results of Section 3 while training is on finite observations and testing is on the (worst case) population loss, under some light-tail properties of the input data samples.

### C.1  RANDOM DESIGN ON TARGET COVARIANCE MATRIX

In Section 3, we consider the case when $\Sigma_T$ is known exactly. This could be viewed as the fixed design setting where training and testing are on the same set of data. In this section, our analysis will include the estimation error on observing finite unlabeled samples of target domain. Let $X_T = [\boldsymbol{x}_1, \cdots \boldsymbol{x}_{n_U}]^\top \in \mathbb{R}^{n_U \times d}$ be $n_U$ (Here $U$ stands for unlabeled data and is used to distinguish from $n_T$ labeled target samples) data samples where $\boldsymbol{x}_i \sim p_T$, and we will use the unlabeled target samples to conduct estimation. We let $\hat{\Sigma}_T = X_T^\top X_T/n_U$.

Let $\hat{L}_{\mathcal{B}}$ to denote the worst case excess risk measured on the observed target samples: $\hat{L}_{\mathcal{B}}(\hat{\boldsymbol{\beta}}) = \max_{\boldsymbol{\beta}^*\in\mathcal{B}} \mathbb{E}_{\boldsymbol{y}_S} \frac{1}{n_U}\|X_T(\hat{\boldsymbol{\beta}}(\boldsymbol{y}_S) - \boldsymbol{\beta}^*)\|^2$. To find the best linear estimator that minimizes $\hat{L}_{\mathcal{B}}$, our

proposed algorithm becomes:

$$\hat{C} \leftarrow \min_{\tau, C} \left\{ r^2 \tau + \frac{\sigma^2}{n_S} \text{Tr}(\hat{\Sigma}_T^{1/2} C \hat{\Sigma}_S^{-1} C^\top \hat{\Sigma}_T^{1/2}) \right\}, \text{ s.t. } (C - I)^\top \hat{\Sigma}_T (C - I) \preceq \tau I. \quad (10)$$

And set $\hat{\boldsymbol{\beta}} = \hat{C} \hat{\Sigma}_S^{-1} X_S^\top \boldsymbol{y}_S / n_S$. We want to show that in spite of the existence of estimation error due to the replacement of $\Sigma_T$ with $\hat{\Sigma}_T$, our generated $\hat{\boldsymbol{\beta}}$ performs well on the worst-case population risk $L_{\mathcal{B}}(\hat{\boldsymbol{\beta}}) := \max_{\boldsymbol{\beta}^*} \mathbb{E}_{\boldsymbol{y}_S} \mathbb{E}_{\boldsymbol{x} \sim p_T} \|\boldsymbol{x}^\top (\hat{\beta}(\boldsymbol{y}_S) - \boldsymbol{\beta}^*)\|^2$ and achieves minimax linear risk (up to constant multiplicative error).

In this section we assume that the data samples is light tail:

**Definition C.1** ($\rho^2$-subgaussian distribution). *We call a distribution $p$ to be $\rho^2$-subgaussian when there exists $\rho > 0$ such that the random vector $\bar{\boldsymbol{x}} \sim \bar{p}$ is $\rho^2$-subgaussian. $\bar{p}$ is the whitening of $p$ such that $\bar{\boldsymbol{x}} \sim \bar{p}$ is equivalent to $\boldsymbol{x} = \Sigma^{1/2} \bar{\boldsymbol{x}} \sim p$, where $\Sigma$ is the covariance matrix of $p$.* [6]

Notice that here the subgaussian parameter is defined on the whitening of the data, and $\rho$ doesn't depend on how large $\|\Sigma\|_{op}$ is.

**Theorem C.2.** *Fix a failure probability $\delta \in (0, 1)$. Suppose target distribution $p_T$ is $\rho^2$-subgaussian, and the sample size in target domain satisfies $n_U \gg \rho^4(d + \log \frac{1}{\delta})$. Let $\hat{\boldsymbol{\beta}} : \boldsymbol{y}_S \rightarrow \hat{C} \hat{\Sigma}_S^{-1} X_S^\top \boldsymbol{y}_S$ where $\hat{C}$ is defined from Eqn. 10. Then with probability at least $1 - \delta$ over the unlabeled samples from target domain, and for each fixed $X_S$ from source domain, our learned estimator $\hat{\boldsymbol{\beta}}(\boldsymbol{y}_S)$ satisfies:*

$$L_{\mathcal{B}}(\hat{\boldsymbol{\beta}}) \leq (1 + O(\sqrt{\frac{\rho^4(d + \log(1/\delta))}{n}})) R_L(\mathcal{B}). \quad (11)$$

*Specifically, when $\Sigma_T$ commutes with $\hat{\Sigma}_S$ or is rank 1, we have:*

$$L_{\mathcal{B}}(\hat{\boldsymbol{\beta}}) \leq (1.25 + O(\sqrt{\frac{\rho^4(d + \log(1/\delta))}{n}})) R_N(\mathcal{B}). \quad (12)$$

Similarly all other results in the paper could be extended to random design with finite samples $X_T$.

*Proof of Theorem C.2.* The proof relies on the two technical claims C.3, C.4.

Let $\hat{\boldsymbol{\beta}}_R$ be the optimal linear estimator on $L_{\mathcal{B}}$, i.e., $L_{\mathcal{B}}(\hat{\boldsymbol{\beta}}_R) = \min_{\boldsymbol{\beta} \text{ linear in } \boldsymbol{y}_S} L_{\mathcal{B}}(\boldsymbol{\beta}) = R_L(\mathcal{B})$.

$$L_{\mathcal{B}}(\hat{\boldsymbol{\beta}}) \leq (1 + O(\sqrt{\frac{\rho^4(d + \log(1/\delta))}{n}})) \hat{L}_{\mathcal{B}}(\hat{\boldsymbol{\beta}}) \quad \text{(Claim C.4)}$$

$$\leq (1 + O(\sqrt{\frac{\rho^4(d + \log(1/\delta))}{n}})) \hat{L}_{\mathcal{B}}(\hat{\boldsymbol{\beta}}_R) \quad \text{(from definition of } \hat{\boldsymbol{\beta}})$$

$$\leq (1 + O(\sqrt{\frac{\rho^4(d + \log(1/\delta))}{n}}))^2 L_{\mathcal{B}}(\hat{\boldsymbol{\beta}}_R) \quad \text{(Claim C.4)}$$

$$\leq (1 + O(\sqrt{\frac{\rho^4(d + \log(1/\delta))}{n}})) L_{\mathcal{B}}(\hat{\boldsymbol{\beta}}) = (1 + O(\sqrt{\frac{\rho^4(d + \log(1/\delta))}{n}})) R_L(\mathcal{B}).$$
$$\text{(from } \frac{\rho^4(d + \log(1/\delta))}{n} \ll 1, \text{ and definition of } \hat{\boldsymbol{\beta}}_R)$$

---

[6] A random vector $\boldsymbol{x}$ is called $\rho^2$-subgaussian if for any fixed unit vector $\boldsymbol{v}$ of the same dimension, the random variable $\boldsymbol{v}^\top \boldsymbol{x}$ is $\rho^2$-subgaussian, i.e., $\mathbb{E}[e^{s \cdot \boldsymbol{v}^\top (\boldsymbol{x} - \mathbb{E}[\boldsymbol{x}])}] \leq e^{s^2 \rho^2 / 2}$ ($\forall s \in \mathbb{R}$).

From Theorem 3.4 we know $R_L(\mathcal{B}) \leq 1.25 R_N(\mathcal{B})$ when $\Sigma_T$ is rank-1 matrix or commute with $\hat{\Sigma}_S$ which further finishes the whole proof. $\qquad\square$

**Claim C.3** (Restated Claim A.6 from Du et al. (2020)). *Fix a failure probability $\delta \in (0, 1)$, and assume $n \gg \rho^4(d + \log(1/\delta))$ [7]. Then with probability at least $1 - \frac{\delta}{10}$ over the inputs $\boldsymbol{x}_1, \ldots, \boldsymbol{x}_n$, if $\boldsymbol{x}_i \sim p$ and $p$ is a $\rho^2$-subgaussian distribution, we have*

$$(1 - O(\sqrt{\frac{\rho^4(d + \log(1/\delta))}{n}}))\Sigma \preceq \frac{1}{n}X^\top X \preceq (1 + O(\sqrt{\frac{\rho^4(d + \log(1/\delta))}{n}}))\Sigma, \qquad (13)$$

*where $\Sigma = \mathbb{E}_{\boldsymbol{x} \sim p}[\boldsymbol{x}\boldsymbol{x}^\top]$.*

With the help of Claim C.3 we directly get:

**Claim C.4.** *Fix a failure probability $\delta \in (0, 1)$, and assume $n_U \gg \rho^4(d + \log(1/\delta))$, $X_T = [\boldsymbol{x}_1, \cdots, \boldsymbol{x}_{n_U}]^\top \in \mathbb{R}^{n_U \times d}$ satisfies $\boldsymbol{x}_i \sim p_T$ where $p_T$ is $\rho^2$-subgaussian. We have for any estimator $\boldsymbol{\beta}$:*

$$(1 - O(\sqrt{\frac{\rho^4(d + \log(1/\delta))}{n_U}}))L_{\mathcal{B}}(\boldsymbol{\beta}) \leq \hat{L}_{\mathcal{B}}(\boldsymbol{\beta}) \leq (1 + O(\sqrt{\frac{\rho^4(d + \log(1/\delta))}{n_U}}))L_{\mathcal{B}}(\boldsymbol{\beta}),$$

*with high probability $1 - \delta/10$ over the random samples $X_T$.*

*Proof of Claim C.4.* Recall

$$\hat{L}_{\mathcal{B}}(\hat{\boldsymbol{\beta}}) = \max_{\boldsymbol{\beta}^* \in \mathcal{B}} \mathbb{E}_{\boldsymbol{y}_S} \frac{1}{n_U}\|X_T(\hat{\boldsymbol{\beta}}(\boldsymbol{y}_S) - \boldsymbol{\beta}^*)\|^2,$$

$$L_{\mathcal{B}}(\hat{\boldsymbol{\beta}}) = \max_{\boldsymbol{\beta}^* \in \mathcal{B}} \mathbb{E}_{\boldsymbol{y}_S} \|\Sigma_T^{1/2}(\hat{\boldsymbol{\beta}}(\boldsymbol{y}_S) - \boldsymbol{\beta}^*)\|^2.$$

Therefore for any estimator $\hat{\boldsymbol{\beta}}$, it satisfies

$$
\begin{aligned}
&L_{\mathcal{B}}(\hat{\boldsymbol{\beta}}) - \hat{L}_{\mathcal{B}}(\hat{\boldsymbol{\beta}}) \\
=&(\hat{\boldsymbol{\beta}}(\boldsymbol{y}_S) - \boldsymbol{\beta}^*)^\top(\Sigma_S - \hat{\Sigma}_S)(\hat{\boldsymbol{\beta}}(\boldsymbol{y}_S) - \boldsymbol{\beta}^*) \\
\lesssim& O(\sqrt{\frac{\rho^4(d + \log(1/\delta))}{n_U}})(\hat{\boldsymbol{\beta}}(\boldsymbol{y}_S) - \boldsymbol{\beta}^*)^\top\Sigma_S(\hat{\boldsymbol{\beta}}(\boldsymbol{y}_S) - \boldsymbol{\beta}^*) \\
=& O(\sqrt{\frac{\rho^4(d + \log(1/\delta))}{n_U}})L_{\mathcal{B}}(\hat{\boldsymbol{\beta}}),
\end{aligned}
$$

which finishes the proof. $\qquad\square$

---

[7]When this is not satisfied the result is still satisfied by replacing $O(\sqrt{\frac{\rho^4(d+\log(1/\delta))}{n}})$ with $O(\max\{\sqrt{\frac{\rho^4(d+\log(1/\delta))}{n}}, \frac{\rho^2(d+\log(1/\delta))}{n}\})$. For cleaner presentation, we assume $n$ is large enough and simplify the results.

## C.2 RANDOM DESIGN ON SOURCE DOMAIN.

In the main text or the subsection above, the worst case excess risk is upper bounded by $1.25 R_N$, which is achieved by best estimator that is using the same set of training data $(X_S, \boldsymbol{y}_S)$. Here we would like to take into consideration the randomness of $X_S$ and compare the worst case excess risk using our estimator with a stronger notion of linear estimator.

For this purpose, we consider estimators that are linear functionals of $\boldsymbol{y}_R := \Sigma_S^{1/2} \boldsymbol{\beta}^* + \boldsymbol{z} \in \mathbb{R}^d, \boldsymbol{z} \sim \mathcal{N}(0, \sigma^2/n_S I_d)$ (this $\sigma^2/n_S$ is the correct scaling since $X_S^\top X_S/n_S$ is comparable to $\Sigma_S$). We consider the minimax linear estimator with $\boldsymbol{y}_R$ and with access to $\Sigma_S$, and we compare our estimator against this oracle linear estimator. This estimator is not computable in practice since $\Sigma_S$ must be estimated, but we will show that our estimator is within an absolute multiplicative constant in minimax risk of the oracle linear estimator.

To recap the notations and setup, let

$$\hat{L}_\mathcal{B}(\hat{\boldsymbol{\beta}}) := \max_{\boldsymbol{\beta}^*} \mathbb{E}_{\boldsymbol{y}_S} \frac{1}{n_U} \|X_T(\hat{\beta}(\boldsymbol{y}_S) - \boldsymbol{\beta}^*)\|^2,$$

$$L_\mathcal{B}(\hat{\boldsymbol{\beta}}) := \max_{\boldsymbol{\beta}^*} \mathbb{E}_{\boldsymbol{y}_S} \mathbb{E}_{\boldsymbol{x} \sim p_T} \|\boldsymbol{x}^\top (\hat{\beta}(\boldsymbol{y}_S) - \boldsymbol{\beta}^*)\|^2,$$

$$L_{\mathcal{B},R}(\hat{\boldsymbol{\beta}}) := \max_{\boldsymbol{\beta}^*} \mathbb{E}_{\boldsymbol{y}_R} \mathbb{E}_{\boldsymbol{x} \sim p_T} \|\boldsymbol{x}^\top (\hat{\beta}(\boldsymbol{y}_R) - \boldsymbol{\beta}^*)\|^2.$$

Our target is to find the best linear estimator using $\hat{L}_\mathcal{B}(\hat{\boldsymbol{\beta}})$ (trained with $X_T$) and prove its performance on the population (worst-case) excess risk $L_\mathcal{B}(\hat{\boldsymbol{\beta}})$ is no much worse compared to the minimax linear risk trained on $\boldsymbol{y}_R$ and $\Sigma_S$.

**Theorem C.5.** *Fix a failure probability $\delta \in (0, 1)$. Suppose both target and source distributions $p_S$ and $p_T$ are $\rho^2$-subgaussian, and the sample sizes in source domain and target domain satisfies $n_S, n_U \gg \rho^4(d + \log \frac{1}{\delta})$. Let $\hat{C}$ be the solution for Eqn.(10), and set $\hat{\boldsymbol{\beta}}(\boldsymbol{y}_S) \leftarrow \hat{C}\hat{\Sigma}_S^{-1} X_S^\top \boldsymbol{y}_S$. Then with probability at least $1 - \delta$ over all the unlabeled samples from target domain and all the labeled samples $X_S$ from source domain, our estimator $\hat{\boldsymbol{\beta}}(\boldsymbol{y}_R)$ yields the worst case expected excess risk that satisfies:*

$$L_\mathcal{B}(\hat{\boldsymbol{\beta}}) \leq \left( 1 + O(\sqrt{\frac{\rho^4(d + \log(1/\delta))}{n_U}}) + O(\sqrt{\frac{\rho^4(d + \log(1/\delta))}{n_T}}) \right) \min_{\boldsymbol{\beta} \ linear \ in \ \boldsymbol{y}_R} L_{R,\mathcal{B}}(\boldsymbol{\beta}).$$

*Proof of Theorem C.5.* For each matrix $C \in \mathbb{R}^{d \times d}$, we first conduct bias-variance decomposition and rewrite each worst-case risk with linear estimator in terms of a matrix $C$. When $\hat{\boldsymbol{\beta}}(\boldsymbol{y}_S) = C\hat{\Sigma}_S^{-1} X_S^\top \boldsymbol{y}_S$, we have:

$$\hat{L}_\mathcal{B}(\hat{\boldsymbol{\beta}}) = \|\hat{\Sigma}_T^{1/2}(C - I)\|_{op}^2 r^2 + \frac{\sigma^2}{n} \text{Tr}(\hat{\Sigma}_T C \hat{\Sigma}_S^{-1} C^\top) =: \hat{l}(C),$$

$$L_\mathcal{B}(\hat{\boldsymbol{\beta}}) = \|\Sigma_T^{1/2}(C - I)\|_{op}^2 r^2 + \frac{\sigma^2}{n} \text{Tr}(\Sigma_T C \hat{\Sigma}_S^{-1} C^\top) =: l(C),$$

Similarly, when $\hat{\boldsymbol{\beta}}_R = C\Sigma_S^{-1/2} \boldsymbol{y}_R$, we have:

$$L_{R,\mathcal{B}}(\hat{\boldsymbol{\beta}}) = \|\Sigma_T^{1/2}(C - I)\|_{op}^2 r^2 + \frac{\sigma^2}{n} \text{Tr}(\Sigma_T C \Sigma_S^{-1} C^\top) =: l_R(C).$$

**Claim C.6.** *Fix a failure probability $\delta \in (0, 1)$, and assume $n_U, n_S \gg \rho^4(d + \log(1/\delta))$, $X_S \in \mathbb{R}^{n_S \times d}$, $X_T \in \mathbb{R}^{n_U \times d}$ are respectively from $p_S$ $p_T$ which are both $\rho^2$-subgaussian. We have for any matrix $C \in \mathbb{R}^{d \times d}$:*

$$(1 - O(\sqrt{\frac{\rho^4(d + \log(1/\delta))}{n_U}}))\hat{l}(C) \le l(C) \le (1 + O(\sqrt{\frac{\rho^4(d + \log(1/\delta))}{n_U}}))\hat{l}(C),$$

*with high probability $1 - \delta/10$ over the random samples $X_T$.*

$$(1 - O(\sqrt{\frac{\rho^4(d + \log(1/\delta))}{n_S}}))l(C) \le l_R(C) \le (1 + O(\sqrt{\frac{\rho^4(d + \log(1/\delta))}{n_S}}))l(C),$$

*with high probability $1 - \delta/10$ over the random samples $X_S$.*

*Proof of Claim C.6.* We omit the proof of the first inequality since it's exactly the same as proof of Claim C.4.

For the second line, we have:

$$\begin{aligned}
l_R(C) - l(C) &= \frac{\sigma^2}{n_S}\mathrm{Tr}(\Sigma_T C(\Sigma_S^{-1} - \hat{\Sigma}_S^{-1})C^\top) \\
&\le O(\sqrt{\frac{\rho^4(d + \log(1/\delta))}{n_S}})\frac{\sigma^2}{n_S}\mathrm{Tr}(\Sigma_T C\hat{\Sigma}_S^{-1}C^\top) \\
&\le O(\sqrt{\frac{\rho^4(d + \log(1/\delta))}{n_S}})l(C).
\end{aligned}$$

Therefore we prove the RHS of the second inequality. The LHS follows with the same proof techniques. $\qquad\square$

Now let $\hat{C}$ be the minimizer for $\hat{l}(C)$, and $C_R$ be the minimizer for $l_R(C)$.

$$\begin{aligned}
l(\hat{C}) \le& (1 + O(\sqrt{\frac{\rho^4(d + \log(1/\delta))}{n_U}}))\hat{l}(\hat{C}) && \text{(w.p. } 1 - \delta/10\text{; due to Claim C.6)} \\
\le& (1 + O(\sqrt{\frac{\rho^4(d + \log(1/\delta))}{n_U}}))\hat{l}(C_R) && \text{(Due to the definition of } \hat{C}\text{)} \\
\le& (1 + O(\sqrt{\frac{\rho^4(d + \log(1/\delta))}{n_U}}))^2 l(C_R) && \text{(w.p. } 1 - \delta/5\text{; due to Claim C.6)} \\
=& (1 + O(\sqrt{\frac{\rho^4(d + \log(1/\delta))}{n_U}}))l(C_R) && \text{(since } n_U \text{ is large enough)} \\
\le& (1 + O(\sqrt{\frac{\rho^4(d + \log(1/\delta))}{n_U}}))(1 + O(\sqrt{\frac{\rho^4(d + \log(1/\delta))}{n_T}}))l_R(C_R) \\
& && \text{(w.p. } 1 - 3\delta/10\text{; due to Claim C.6)}
\end{aligned}$$

$$= \left(1 + O(\sqrt{\frac{\rho^4(d + \log(1/\delta))}{n_U}}) + O(\sqrt{\frac{\rho^4(d + \log(1/\delta))}{n_T}})\right) \min_C l_R(C).$$

This finishes the proof. □

## D MORE EMPIRICAL RESULTS

We include some more empirical studies. In the main text our results have small noise. Here we show some more results with larger noise, and also the case with varied eigenspace. For the following results, we use $\sigma = 10$ and $r = 0.2\sqrt{d}$. Other meta data remains the same as presented in the main text. Figure 2 (a)(b) show similar phenomenon as the small noise setting presented in

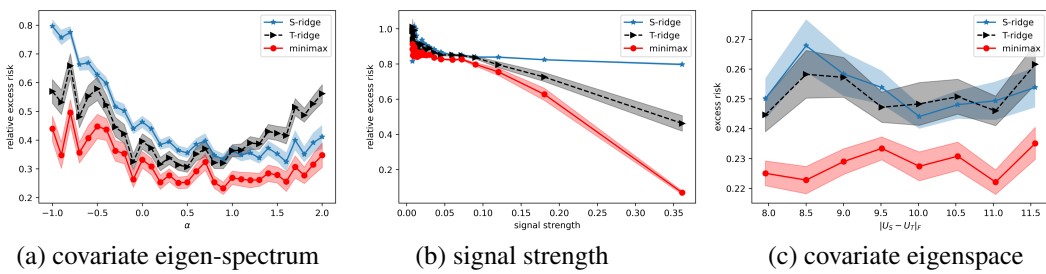

(a) covariate eigen-spectrum     (b) signal strength     (c) covariate eigenspace

Figure 2: (a): The x-axis $\alpha$ defines the spread of eigen-spectrum of $\Sigma_S$: $s_i \propto 1/i^\alpha, t_i \propto 1/i$. (b) x-axis is the normalized value of signal strength: $\|\Sigma_T \beta^*\|/r$. (c) X-axis is the covariate shift due to eigenspace shift measured by $\|U_S - U_T\|_F$.

the main text. From Figure 2 (c) we see no particular relationship between the performance of each algorithm with eigenspace shift.

### D.1 EXPERIMENTS WITH APPROXIMATION ERROR

Finally, we conduct empirical studies with nonlinear models.

**Setup.** We choose $n_S = 2000, d = 50$. Let $X_S \in \mathbb{R}^{2000 \times 50}$ be generated randomly under Gaussian distribution $\mathcal{N}(0, \Sigma_S)$. We also generate a small validation dataset from target domain: $X_{CV} \in \mathbb{R}^{500 \times 50}$, sampled from $\mathcal{N}(0, \Sigma_T)$, $y_{CV} = f^*(X_{CV}) + z_{CV}$, with $z_{CV} \sim \mathcal{N}(0, \sigma^2 I)$. We choose $\lambda_i(\Sigma_S) \propto i, \lambda_i(\Sigma_T) \propto 1/i$, and the eigenspace for both $\Sigma_S$ and $\Sigma_T$ are random orthonormal matrices. ($\|\Sigma_S\|_F^2 = \|\Sigma_T\|_F^2 = d$.) The ground truth model is a one-hidden-layer ReLU network: $f^*(x) = 1/d a^\top (Wx)_+$, where $W$ and $a$ are randomly generated from standard Gaussian distribution. We observe noisy labels: $y_S = f^*(x) + z$, where $z_i \sim \mathcal{N}(0, \sigma^2)$.

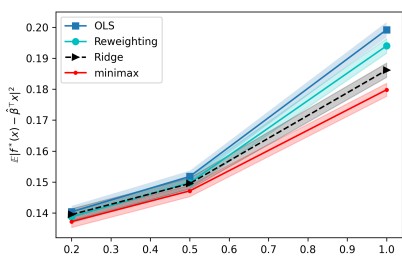

Figure 3: The x-axis is noise level $\sigma$ and y-axis is the excess risk (with approximation error).

**Estimating weights $p_T(\boldsymbol{x})/p_S(\boldsymbol{x})$.** Since the generated data samples are Gaussian, the absolute weights for $p_T(\boldsymbol{x})/p_S(\boldsymbol{x}) = \sqrt{\frac{|\Sigma_S|}{|\Sigma_T|}} \exp(\frac{1}{2}\boldsymbol{x}^\top(\Sigma_S^{-1} - \Sigma_T^{-1})\boldsymbol{x})$. However, this absolute value has an exponential factor and can amplify the noise level. Meanwhile, when one multiplies both $X_S, \boldsymbol{y}_S$ by 10, the ground truth $\beta$ doesn't change but the absolute value for $p_T(\boldsymbol{x})/p_S(\boldsymbol{x})$ will change drastically. This discrepancy highlights the importance of relative magnitudes (among samples) instead of the absolute value Kanamori et al. (2009).

To obtain a relative score, we first estimate the absolute value of $p_T(\boldsymbol{x})/p_S(\boldsymbol{x})$ by $l(\boldsymbol{x}) := \boldsymbol{x}^\top(\hat{\Sigma}_S^{-1} - \hat{\Sigma}_T^{-1})\boldsymbol{x}$. We then uniformly assign the weight for each sample by 10 discrete values $1, 2, 3 \cdots 10$ based on their scoring $l(\boldsymbol{x})$ and then rescale the reweighting vector properly.

We implement our method (Eqn. 4) using the estimated weights as above. Refer to Figure 3 for the results. The baselines we choose are ordinary least square ("OLS" in Figure (3)), ridge regression (Legend is "Ridge") and classic weighted least square Kanamori et al. (2009) (Legend is "Reweighting"; $\hat{\boldsymbol{\beta}}_{\text{LS}}$ in our main text). For both ridge regression and our methods, we tune hyperparameters through cross-validation. All results are presented from 40 runs where the randomness comes from $f^*$ and the eigenspaces of $\Sigma_S, \Sigma_T$.

