# OpenReview forum: "Near-Optimal Linear Regression under Distribution Shift"
_ICLR.cc/2021/Conference — Reject_

### Official Review · AnonReviewer3 · 2020-10-25

**Rating:** 6
**Confidence:** 2

**Review:**

This paper studies the problem of domain adaptation for linear regression problem.

1. First the authors consider a well-specified model where only the distribution of the covariates changes. For this case the authors propose a linear estimator, and show that under assumptions which guarantees the reduction of the problem to Gaussian sequence, this estimator is within a constant factor from the minimax risk of non-linear estimators.
2. Then, the authors turn their attention to the miss-specified case of the above model. They again develop an estimator whose maximal risk is shown to match the minimax one (for linear estimators).
3. The authors consider a well-specified model allowing the vector $\beta^*$ to be changes from source to target. Similar results as for the previous case are derived.
4. The authors conduct simulation study comparing their estimators with the ridge regression.


pros: the developed estimators are theoretically grounded

cons: after several readings I still cannot understand why this work is on domain adaptation. Since the authors consider the problem of linear regression with fixed design, the domain adaptation part simply reduces to the re-weighting of the norm.

I am leaning towards rejection, but I am willing to raise the score if the authors are able to address my questions below.

Questions:
1. The authors work with the fixed design setup, that is, the expectation in the risk is taken only w.r.t. the signal $y_S$. What is the meaning of source and target data when the designs are fixed? For instance, in section 3.3. can we simply pull together $y_S, y_T$ and $X_S, X_T$ and apply the result of previous section?
2. The authors should clarify how their first contribution is different from that of **Blaker, Minimax estimation in linear regression under restrictions, 2000** (Theorem 1 and the discussion after it).
3. In Theorem 3.4 there is a condition that $\Sigma_S, \Sigma_T$ should commute. Isn't it $\hat{\Sigma}_S$ instead? If not, what is the reason to introduce this assumption if the design is fixed?
4. If the covariance matrix on the target is unknown, we would have to estimate it using unlabelled sample. Did you investigate a theoretical or empirical impact on the problem?

Suggestion: It might be useful to write few lines explaining why the excess risk on the target reduces to the 2-norm re-weighted by the target covariance.


After rebuttal: They authors have addressed my questions. I hope that the authors will carefully review classical literature on weighted minimax linear estimator in their final version. I raise my score to marginally above acceptance threshold, since I am still not convinced in the importance of domain adaptation for fixed design regression.

---

> ### Author Response · Authors · 2020-11-14
> **Response to Reviewer 3**
>
> Thank you for the valuable feedback. We address your questions below.
>
> "Don't understand why this work is on domain adaptation": First we would like to emphasize that the settings we consider are standard in domain adaptation, where we want to use labeled data from a source task to solve a target task from which few or no labeled data are available. We consider multiple types of distribution shifts including covariate shift and model shift. The fixed design assumption is not crucial and our results extend to random design (see below). Although for linear regression the distribution shift is characterized by the covariance change, this doesn't trivialize the problem. The key message is: when the source and target data have different covariance matrices, we should regularize less on the directions where target distribution has higher signal strength (see discussions below Theorem 3.2).
>
> 1. "What is the meaning of source and target data when the designs are fixed": We would like to clarify that our algorithms work for random design and our empirical verification is also with randomly generated source and target data. The fixed design only serves as a standard setting in order to prove the minimax lower bound (i.e., in the definition of $R_L$ and $R_N$, the expectation is only over $y$ but not $\mathbf{x}$). Typically lower bounds in the fixed design setting can be extended to random design since random design is a harder statistical problem (there is more randomness/uncertainty in random design and thus the same lower bound applies) and can be made formal via Le Cam distance. It is also straightforward to extend our upper bound analysis to random design using covariance concentration, as we can estimate the covariance matrix $\Sigma_T$ using unlabeled data. For instance, when input is sub-Gaussian (light-tail and thus easily concentrated), it is ensured that the empirical covariance matrix $\hat\Sigma_T$ satisfies $0.9 \hat \Sigma_T \preceq \Sigma_T\preceq 1.1\hat \Sigma_T $ with $\Omega(d)$ samples (see e.g. Claim A.6 in [1]). Therefore, our algorithm in the random design setting attains the same upper bound rate. We will make this point clearer in the revised version.
> "In section 3.3. can we simply pull together
> $y_S,y_T$ and $X_S,X_T$ and apply the result of the previous section":
> Section 3.3 is a strict generalization of Section 3.1 when there are observed labeled data from the target task. The arguments follow the same recipe as Section 3.1, with similar conclusions. We chose to present Section 3.1 first since the arguments there are easier to follow.
>
> 2. "How the first contribution differs from Blaker 2000": First, [Blaker 2000] didn't consider distribution shift, and we propose the first algorithm that achieves minimax risk (within constant factor) under distribution shift. Second, for the general non-commutative case we derive the optimal linear estimator (Eq. (3)) which is very different from [Blaker 2000].
>
> 3. Yes, it should be $\hat \Sigma_S$. Thank you for pointing this out.
>
> 4. "Estimate covariance matrix for target domain": Yes, this can be handled theoretically and is indeed the setting used in our experiments. See answer to Q1.
>
> [1] Du, Simon S. et al. "Few-shot learning via learning the representation, provably"

---

### Official Review · AnonReviewer1 · 2020-11-02
**Paper that has good contributions for analyzing covariate shift. Algorithms and theory is supported by experiments.**

**Rating:** 6
**Confidence:** 3

**Review:**

## Summary
The authors propose a near minimax optimal linear estimator under distribution shift. They have estimators for when there is a covariate shift (i.e. the underlying data distribution) or model shift (i.e. the distribution of the label given the features of the data).

## Strengths
* The authors provide bounds for both data coming from linear and non-linear generative models
* The authors analyze model shift in addition to covariate shift
* Experiments showing the results on simulated data

## Weaknesses
* No experiments for the non-linear data generation model

Update: after reading the arguments and comments from my fellow reviewers, I was convinced by their reasoning that the paper did not merit my original high score. I would like to therefore change my score to a 6.

---

> ### Author Response · Authors · 2020-11-14
> **Response to Reviewer 1**
>
> Thank you very much for your encouraging review!
>
> We are working on simulations for the non-linear data generation setting.

---

### Official Review · AnonReviewer5 · 2020-11-09
**Good paper on minimax linear estimators under distributional shift, with some lack clarity.**

**Rating:** 6
**Confidence:** 4

**Review:**

### Summary
This paper derives estimators for regression under additive Gaussian noise and distributional shift (mostly covariate and also some model shift) that enjoy minimax properties.

### Strengths
+ Sample complexity bounds, let alone matching lower bounds, are only recently being proposed for various realistic frameworks with distributional shift. This paper is thus timely and well-motivated.
+ The presentation is clear, and most analysis steps are well-delineated.
+ The insight that linear sufficient statistics could achieve minimax rates is valuable and the comparison to ridge regression, especially when accounting for the target, is enlightening.

### Weaknesses
- Some of the assumptions are rather strong. Assuming that the target covariance $\Sigma_T$ is known is somewhat acceptable. But when dealing with model shift, the paper assumes knowing the entire marginal distributions in the source $p_S$ and target $p_T$ (to get the importance weights $\mathbf{w}$), which effectively sidesteps the covariate shift entirely.
- ~~*[Major]* In the proof of Claim 3.1 (page 12), a key step appears wrong to me: namely the step that says “Cross term vanishes”. While it’s true that  $W^\intercal X_S = 0$, the outer product $X_S W^\intercal$ is not, and neither is $X_S A_1^\intercal A_2 W^\intercal$ that appears in the cross product and which could contribute negatively with varying $A_2$. Now it could be that this claim can be patched, which is why I judge the rest of the paper by assuming it’s true. I do the same for Claims 3.6 and 3.8, which suffer from the same issue. (Claim 3.8 just relies on the previous two, and Claim 3.6 omits the same  cross term on the third line of page 16, which invalidates the argument on the first two lines of the paragraph that follows “it only appears in the third and last non-negative terms”.)~~

### Comments
* Throughout $X$ is assumed to be centered, I believe. Best to state this explicitly.
* The definition of a sufficient statistic in footnote 2 is not the right one to use, since we cannot talk of independence when $\beta^\star$ is an unknown parameter. (A corresponding justification of (1) as sufficient statistic should then be given, using sufficiency of optimal estimation.)
* Typos:
   - the dimension of $X_S$ is $n_S \times d$ (subscript $S$ missing)
   - in the last equation of page 4 I believe the reference should be to Claim 3.1
   - in section 4 $\bar\beta_T$ needs a division by $n_T$
   - in the proof of Claim 3.1, “This is to show […] minimax *linear* estimator”
   - in the proof of Theorem 3.2, second line, divide $\mathbf{y}_T$ by $n_S$ and insert a missing $=$ sign before $\theta^\star_T$; in the third line change the subscript of $\Sigma$ to $T$ in the definition of $\Theta$; generally make sure the bolding convention is kept in this (otherwise rather nice) proof

### Overall

Minimax estimators for distributional shift can be very useful. But the paper has some lack of clarity and potential flaws, which is the only reason behind my current rating.

_[Edit: The authors clarified that the flaw I was talking about isn't there. I updated my review accordingly. The score is based on the fact that the paper can use some polish to improve clarity and the fact that some of the assumptions, even if traditional, are too strong (in essence, sample complexity bound will become meaningless if what is assumed known itself is too sample-intensive to approximate).]_

---

> ### Author Response · Authors · 2020-11-14
> **Response to Reviewer 5**
>
> Thank you for the insightful suggestions and for pointing out the unclear parts in our derivations. We would like to emphasize that there is no mistake in our derivations. Please see our response below. We have updated the paper to avoid future confusion.
>
> 1. "[Major] Proof of Claim 3.1 cross-term": We would like to emphasize that our derivations are correct.
> In the proof of Claim 3.1, the cross term in question is indeed zero due to the cyclic property of matrix trace:
> $\mathbb{E} \langle \Sigma_T^{1/2}A_1X_S^\top \mathbf{z}, \Sigma_T^{1/2}A_2W^\top \mathbf{z} \rangle
> =\mathbb{E}[\mathrm{Tr}(\Sigma_T^{1/2}A_2 W^\top \mathbf{z} \mathbf{z}^\top X_S A_1^\top \Sigma_T^{1/2})]$
> $=\mathrm{Tr}(\Sigma_T^{1/2}A_2 W^\top \mathbb{E}[\mathbf{z} \mathbf{z}^\top] X_S A_1^\top \Sigma_T^{1/2})$
> $= \sigma^2 \mathrm{Tr}(\Sigma_T^{1/2}A_2 (W^\top X_S) A_1^\top \Sigma_T^{1/2})=0. $
> We have added this calculation to the end of page 12. Similarly this argument applies to Claims 3.6 and 3.8. We hope this resolves your concern about the correctness.
>
> 2. "Assuming that the target covariance $\Sigma_T$ is known": Regarding  $\Sigma_T$, we could use unlabeled data to calculate the empirical covariance $\hat \Sigma_T= X_T^\top X_T/n_T$ and implement the algorithms using $\hat \Sigma_T$. In the theoretical analysis, we considered the setting of fixed design, which is conventional in order to show minimax lower bounds, and so we did not distinguish between $\Sigma_T$ and $\hat \Sigma_T$.
> The minimax lower bound still holds in the random design setting since fixed design is an easier statistical problem than random design, a consequence of Le Cam distance lemma.
> For the upper bound analysis in the random design setting, we need to additionally ensure that $\hat \Sigma_T$ concentrates around $\Sigma_T$.
> This is typically satisfied when the input satisfies a certain standard light-tail assumption. For example, when the input is sub-Gaussian, it satisfies that $0.9 \hat \Sigma_T \preceq \Sigma_T\preceq 1.1\hat \Sigma_T $ when there are $\Omega(d)$ samples (see e.g. Claim A.6 in [1]). Therefore again we can achieve near-optimal (within constant factor) minimax risk when the algorithm uses $\hat \Sigma_T$.
>
>
> 3. "The paper assumes to know $p_T$ and $p_S$":  Regarding the reweighing algorithm with $p_T/p_S$, we would like to clarify our claims and contributions. We agree that estimating $p_T$ and $p_S$ is a long-standing and important problem in the domain adaptation with many proposed algorithms to estimate importance weights. As we mentioned in the related work, the reweighing algorithm has been used for years and therefore it is not our contribution; neither is estimating $p_T$ or $p_S$.
> However, knowing the importance weights does not sidestep the issue of covariate shift. Importance weighing greatly inflates the variance of the estimator, and our contribution in Section 3.2 is designing an optimal algorithm to cope with the inflated variance.
>
>
> [1] Du, Simon S. et al. "Few-shot learning via learning the representation, provably"

---

### Author Response · Authors · 2020-11-24
**Summary on final revisions**

We thank all the reviewers for their detailed feedback and insightful suggestions. We have revised our paper to address all the issues pointed out by the reviewers. Specifically, we have done the following modifications.

1. We add theoretical analysis when we estimate $\Sigma_T$ using unlabeled data from the target domain. We show that under some standard light-tail assumptions, with sufficient samples $(\gg d)$, our estimator is still within a constant factor of minimax risk. See Appendix C.1.

2. We compare our estimator with a stronger linear estimator that has unlimited access to $p_S$ and show it is within a constant of the stronger linear estimator (when $n_S\gg d$).
See Appendix C.2.

3. We add some simulations with approximation error (when the ground truth model is nonlinear). In the simulations, we estimate the weights by empirical covariance matrices and show better results than ridge regression and other baselines.

4. We revise the typos and add some omitted statements/calculations.

We sincerely hope reviewers \#5 and reviewer \#3 will reconsider their scores since we believe we have addressed all the issues you have raised.

R5: "Minimax estimators for distributional shift can be very useful. But the paper has some lack of clarity and potential flaws, which is the only reason behind my current rating." We have added explanations for our calculations in the response as well as in the paper.

R3: "I am leaning towards rejection, but I am willing to raise the score if the authors are able to address my questions below."  We believe we have addressed all the questions you raised.

If you have further questions, we hope you could mention them before the discussion phase ends.

---

### Decision · Program_Chairs · 2021-01-07
**Final Decision**

**Decision:**

Reject

**Comment:**


This paper derives estimators and minimax guarantees for regression under additive Gaussian noise and distributional shift
Despite, some merits raised, limitations on too strong assumptions (knowing the entire marginal distributions, sample complexity bound that could become meaningless if what is assumed known itself is too sample-intensive to approximate) were considered important drawbacks by the reviewers.
Last but not least, the importance of domain adaptation for fixed design regression was questioned and the answer to that point was not convincing enough.